# Two DOT1 enzymes cooperatively mediate efficient ubiquitin-independent histone H3 lysine 76 tri-methylation in kinetoplastids

Victoria S. Frisbie [1,5], Hideharu Hashimoto[1,5], Yixuan Xie [2,3], Francisca N. De Luna Vitorino [2], Josue Baeza[3], Tam Nguyen[1], Zhangerjiao Yuan[1], Janna Kiselar[4], Benjamin A. Garcia[2,3] & Erik W. Debler [1] ✉

In higher eukaryotes, a single DOT1 histone H3 lysine 79 (H3K79) methyltransferase processively produces H3K79me2/me3 through histone H2B mono-ubiquitin interaction, while the kinetoplastid *Trypanosoma brucei* di-methyltransferase DOT1A and tri-methyltransferase DOT1B efficiently methylate the homologous H3K76 without H2B mono-ubiquitination. Based on structural and biochemical analyses of DOT1A, we identify key residues in the methyltransferase motifs VI and X for efficient ubiquitin-independent H3K76 methylation in kinetoplastids. Substitution of a basic to an acidic residue within motif VI ($Gx_6\underline{K}$) is essential to stabilize the DOT1A enzyme-substrate complex, while substitution of the motif X sequence VYGE by CAKS renders a rigid active-site loop flexible, implying a distinct mechanism of substrate recognition. We further reveal distinct methylation kinetics and substrate preferences of DOT1A (H3K76me0) and DOT1B (DOT1A products H3K76me1/me2) in vitro, determined by a Ser and Ala residue within motif IV, respectively, enabling DOT1A and DOT1B to mediate efficient H3K76 tri-methylation non-processively but cooperatively, and suggesting why kinetoplastids have evolved two DOT1 enzymes.

In eukaryotes, the nucleosome is the fundamental unit of chromatin and regulates DNA-templated events by histone modification including methylation, acetylation, phosphorylation, ubiquitination, and SUMOylation[1,2]. Histone residues are modified during transcription, replication, and repair to recruit modification-specific proteins and enzymes[3]. In human and yeast, the DOT1 methyltransferase is solely responsible for histone H3 lysine 79 (H3K79) methylation. H3K79me2/me3 are co-localized with H3K4me3, H4K16 acetylation (H4K16ac), and H2B mono-ubiquitination (H2Bub1) at active transcription start sites[4,5].

The ~350-residue DOT1 methyltransferase domain of *Saccharomyces cerevisiae* Dot1p, human DOT1L[6], *Caenorhabditis elegans* DOT1, *Drosophila melanogaster* DOT1, and *Anopheles gambiae* DOT1 is highly conserved. The DOT1 methyltransferase domain is comprised of an N-terminal helical domain and a C-terminal methyltransferase core domain[7,8]. The N-terminal helical domain slightly diverged among them[8] and interacts with positively-charged residues within the histone H4 N-terminal tail[9,10]. The histone H4 N-terminal tail interaction is essential for methyltransferase activity by stabilizing the projected lysine toward the methyl donor *S*-adenosyl-L-methionine (AdoMet) by hydrogen-bond interactions between the guanidino group of arginine H4R19 and main-chain carbonyl oxygens of the histone H3 L1 loop containing the target lysine[9–11]. Within the C-terminal methyltransferase

[1]Department of Biochemistry and Molecular Biology, Thomas Jefferson University, Philadelphia, PA, USA. [2]Department of Biochemistry and Molecular Biophysics, Washington University School of Medicine, St. Louis, MO, USA. [3]Epigenetics Institute, Department of Biochemistry and Biophysics, Perelman School of Medicine at the University of Pennsylvania, Philadelphia, PA, USA. [4]Case Center for Proteomics and Bioinformatics, Department of Nutrition, Case Western Reserve University, School of Medicine, Cleveland, OH, USA. [5]These authors contributed equally: Victoria S. Frisbie, Hideharu Hashimoto. ✉e-mail: Erik.Debler@jefferson.edu

core domain, seven conserved methyltransferase motifs are identified and invariant among them[8]: Motifs X (VYGE), I (DLGSGxG), II (GxE), III (F(V/L)), IV (NNF(L/A)F), VI (Gx$_6$K), and VIII (VSWT) (nomenclature according to ref. 12, where x denotes any amino acid residue). Motifs X (Val135 and Gly137, residue numbering refers to human DOT1L), IV (Phe243 and Ala244), and VIII (Trp305) form a hydrophobic lysine-binding channel, while residues in motifs X, I, II, and III recognize AdoMet and its product S-adenosyl-L-homocysteine (AdoHcy)[9,10].

Due to AdoMet/AdoHcy being buried in the active site of DOT1 enzymes[7–10] and the conserved motif X-containing loop GH blocking diffusion of AdoMet/AdoHcy from the active-form DOT1-nucleosome complex[7–10], DOT1 methyltransferases must disengage from their product nucleosome after one turnover to exchange AdoHcy to AdoMet for subsequent methylation step(s)[8,13,14]. Interactions of the arginine anchor residues of DOT1 with the acidic patch of the nucleosome and of DNA with the DNA-binding region outside the methyltransferase domain are essential for DOT1 methyltransferase activity[7–9]. For di- or tri-methylation, interaction between mono-ubiquitin at H2B (H2Bub1) and the DOT1L/Dot1p C-terminal ubiquitin-binding helix is required to tether the DOT1 enzyme to the targeting nucleosome[9,10,14,15], resulting in H2Bub1 co-localization with H3K79me2/me3 in the genome[16].

DOT1 enzymes are highly conserved across metazoans and fungi, but not in the early-branched eukaryotic class of kinetoplastids including *Trypanosoma brucei* subsp., *Trypanosoma cruzi*, and *Leishmania* spp. In kinetoplastids, three DOT1-like methyltransferases have been identified, two of which (*T. brucei* DOT1A and DOT1B) have demonstrated methyltransferase activity toward H3K76 (corresponding to H3K79 in human and yeast) in vivo and in vitro[17–19]. Both DOT1A and DOT1B represent minimal enzymes with less than 300 residues, comprised of a significantly diverged N-terminal domain (magenta in Fig. 1a, b) and a conserved C-terminal methyltransferase core domain (dark blue) that contains canonical and non-canonical methyltransferase motifs (Fig. 1a, b). Motifs I, II, III are canonical, while motifs IV, VI, and VIII are non-canonical with substitutions found in key residues. Markedly, the motif X sequence is completely missing in DOT1A and DOT1B and replaced with the sequence CAKS that is invariant among kinetoplastids. Kinetoplastid DOT1 enzymes do not use ubiquitin interaction because they lack a ubiquitin-binding helix (green in Dot1p and DOT1L in Fig. 1a, b)[20,21].

Histone sequences are one of the most conserved ones among model eukaryotes, but kinetoplastid canonical histone sequences, unlike protozoa *Naegleria* or *Plasmodium* canonical histone sequences[22,23], are highly diverged with respect to the human histone sequences[24–26], with sequence identity ranging from 40 to 60% and sequence similarity ranging from 65 to 80%[17,27,28]. Histone H2B ubiquitination is not observed in kinetoplastids due to the substitution of the ubiquitination site (H2B lysine 120 in humans and H2B lysine 123 in yeast) to a histidine[29,30]. Due to the highly diverged histone sequences, histone modifications are quite different[24], and it is difficult to assess whether the Lys-to-His substitution in *T. brucei* H2B impacts histone modification at this site and potential cross-talk with other histone modifications. Consistent with an apparent lack of cross-talk with H2B ubiquitination, H3K76 methylation is broadly distributed throughout the *T. brucei* genome, with the majority of H3K76 being tri-methylated[31]. Unlike human and yeast, in which 1–2% of H3K79me2/3 is co-localized with H2B mono-ubiquitination at K120 in human or K123 in yeast, H3K4me3, and H4K10ac[32,33], the genomic localization of *T. brucei* H3K76me3 is independent of H2A.Z, H4K10ac, or H3K4me3[31,33]. These findings suggest that kinetoplastid DOT1 enzymes have evolved unique substrate recognition mechanisms without H2Bub1 interaction.

Both DOT1A and DOT1B mRNA are constantly detected throughout the cell cycle and are slightly increased in G2 phase of the cell cycle based on mRNA-seq analysis[34], while their protein levels have not been reported yet[18]. H3K76 methylation appears to be cell cycle regulated, as H3K76me1 is mainly detected in G2 and M, while H3K76me2 is only detected at the G2/M transition and during cytokinesis[35,36]. By contrast, H3K76me3 is constitutively detected, and newly incorporated histone H3K76 residues are thought to remain unmodified (H3K76me0) until the G2/M transition[13,37,38]. The H3K76 di-methyltransferase DOT1A is encoded by an essential gene and involved in cell cycle progression[17,18], as *DOT1A* knock-down decreases DNA content, whereas DOT1A over-expression induces DOT1A methyltransferase activity-dependent aneuploid DNA content (>4n). By contrast, the gene encoding the H3K76 tri-methyltransferase DOT1B is non-essential and its knock-out does not induce a cell cycle defect, as loss of *DOT1B* only disrupts monogenic expression of the major cell surface antigen termed Variant Surface Glycoprotein (VSG) and differentiation[18,35,39]. Although DOT1B can generate H3K76me3 from me0 in vitro, it cannot compensate for H3K76 di-methyltransferase DOT1A-RNAi defects in vivo[18]. Various explanations to this conundrum have been discussed[18], but a molecular basis for this observation has remained elusive.

Here we have uncovered distinct kinetics of *T. brucei* DOT1A and DOT1B H3K76 methylation using mass spectrometry (MS) and, based on crystal structure analysis of *T. brucei* DOT1A and mutagenesis, reveal key residues for ubiquitin-independent DOT1 substrate recognition distinct from that in humans and yeast. Our data suggest a mechanism of how kinetoplastid DOT1A and DOT1B cooperatively maintain efficient H3K76 tri-methylation without ubiquitination in vivo and offer a rationale for why DOT1B depletion cannot compensate *DOT1A* knockdown.

## Results

### Generation of stable, active DOT1A and DOT1B enzymes

Previous studies on *T. brucei* DOT1A and DOT1B used GST- or MBP-tagged full-length proteins expressed in *E. coli*, but the full-length enzymes were unstable after GST- or MBP-tag cleavage[18,19]. Among kinetoplastid DOT1 enzymes, the first ~45 residues of DOT1A and the first ~30 residues of DOT1B enzymes are non-conserved, while the remaining C-terminal portions of DOT1A (residues 46–295 in *T. brucei*) and DOT1B (residues 29–275 in *T. brucei*) are highly conserved, delineating the boundaries of the methyltransferase core domain in *T. brucei* DOT1A and DOT1B (Fig. 1). To validate these domain boundaries in *T. brucei* DOT1A, we expressed hexahistidine-tagged full-length DOT1A in Sf9 insect cells and purified it as an active methyltransferase, which showed slight protein degradation (Fig. 2a, b and Supplementary Fig. 1). Limited proteolysis by chymotrypsin and elastase generated a stable, ~27-kDa fragment (Supplementary Fig. 2). Based on this result and the domain boundary prediction, we determined the DOT1A fragment to comprise residues 43 to 295 (Δ42-DOT1A) (Fig. 1). Indeed, the hexahistidine-tagged Δ42-DOT1A was stably expressed in *E. coli* and retained methyltransferase activity (Fig. 2a, b). Due to instability of the hexahistidine-tagged full-length DOT1B expressed in Sf9 cells, we expressed a DOT1B construct termed Δ27-DOT1B (residues 28–275 of DOT1B) based on the Δ42-DOT1A construct in *E. coli* (Figs. 1 and 2a). Both Δ42-DOT1A and Δ27-DOT1B were verified as active and stable H3K76 methyltransferases (Fig. 2b).

### The non-conserved N-terminal region of DOT1A weakly binds DNA but is dispensable for catalytic activity

The non-conserved N-terminal regions of kinetoplastid DOT1 enzymes are enriched in positively-charged Arg/Lys/His residues, indicative of DNA binding. Therefore, we measured DOT1A DNA binding affinity by a fluorescence polarization assay and found that full-length DOT1A has DNA-binding affinity with a $K_D$ value of $0.9 \pm 0.2$ μM, while DNA binding of Δ42-DOT1A was not detectable (Fig. 2c). The DNA-binding affinity of full-length DOT1A is ~10 times lower than that of human DOT1L ($K_D$ of $0.11 \pm 0.02$ μM for DOT1L comprising residues 1–420 including the DNA-binding region, Fig. 2c). Interestingly, human DOT1L requires the DNA-binding region for H3K79 methyltransferase activity, whereas the DOT1A DNA-binding region is dispensable for H3K76 methyltransferase activity (Fig. 2b, c)[6–8]. The non-conserved

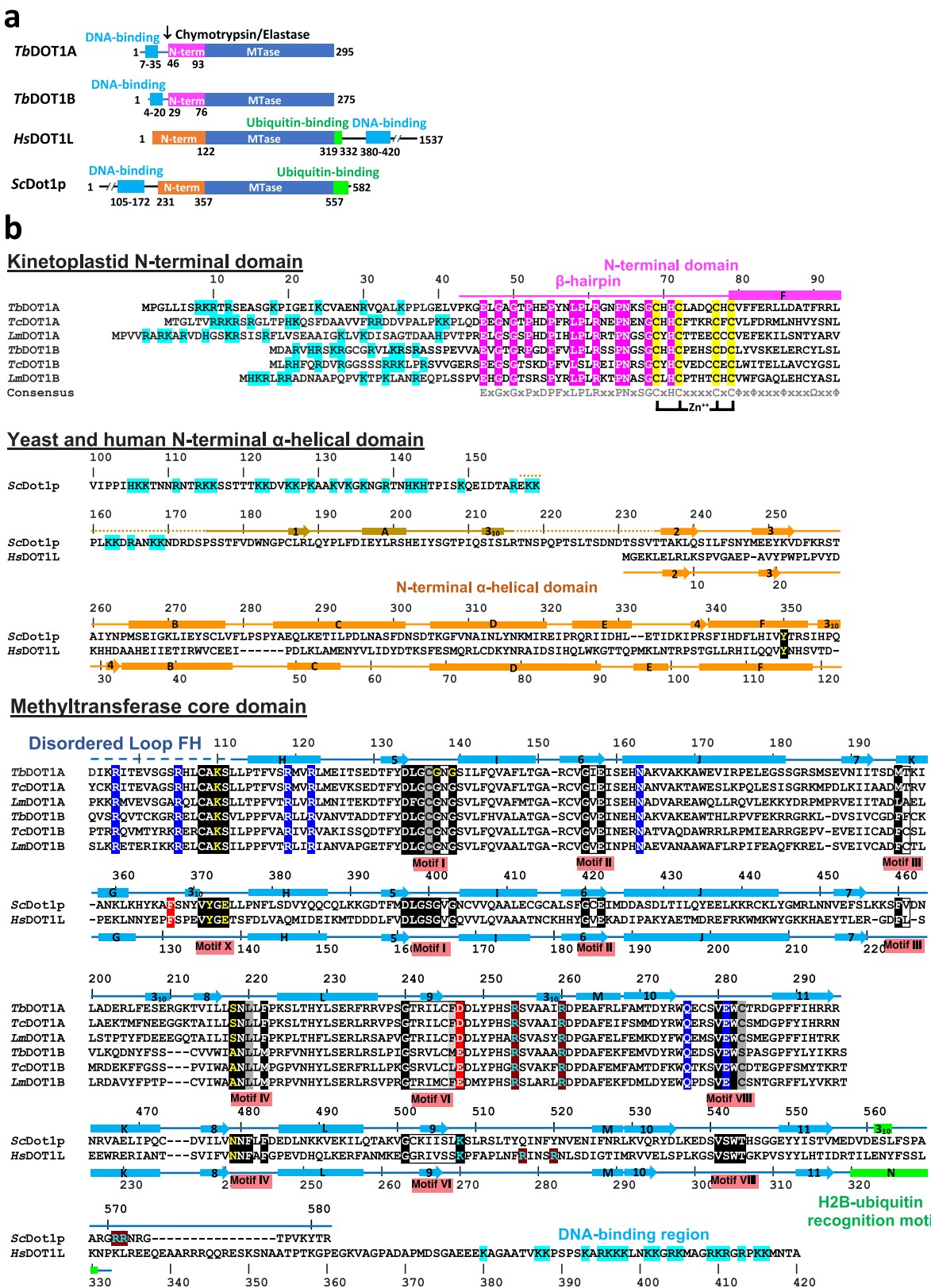

**Fig. 1 | Domain architecture and sequence alignment of DOT1 enzymes.**
**a** Schematic representation of *T. brucei* DOT1A, *T. brucei* DOT1B, human DOT1L, and *S. cerevisiae* Dot1p. The Lys/Arg-rich DNA-binding region (cyan), N-terminal domain (magenta in *T. brucei* DOT1A and DOT1B or orange in *S. cerevisiae* and human), methyltransferase (MTase) core domain (blue), H2B-ubiquitin recognition motif (green), and chymotrypsin/elastase cleavage site are shown. **b** Structure-based sequence alignment of DOT1A and DOT1B of *T. brucei* (TriTrypDB: Tb427_080022000 and Tb427_010005600), *T. cruzi* (TriTrypDB: TcCLB.511391.110 and TcCLB.511417.70), *L. major* (TriTrypDB: LmjF07.0025 and LmjF20.0030), *S. cerevisiae* Dot1p (UniProt: Q04089), and human DOT1L (UniProt: Q8TEK3).

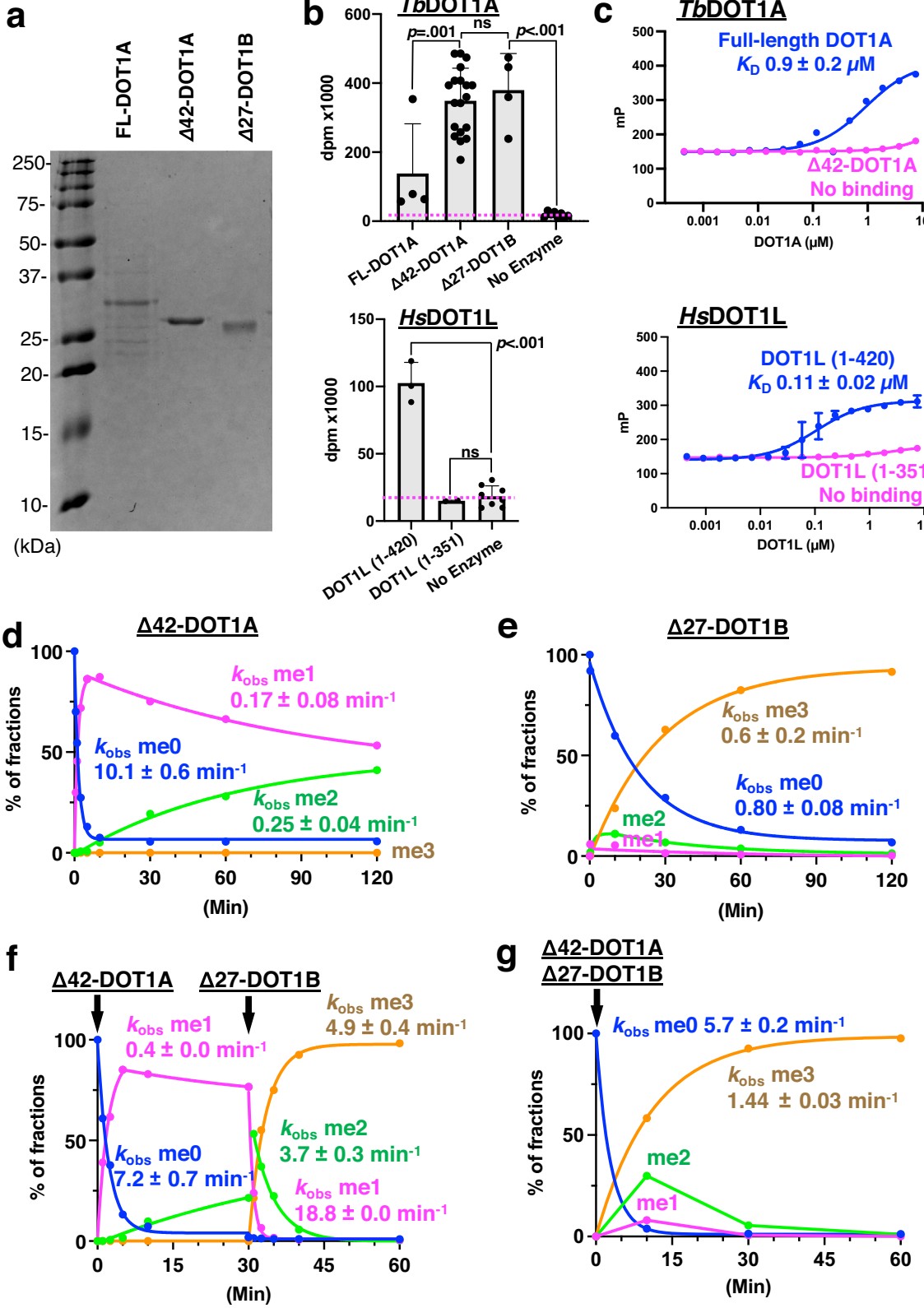

DNA-binding region is not predicted to form a stable domain structure consistent with our limited proteolysis results (Supplementary Fig. 2). Akin to the DNA-binding regions of DOT1L and Dot1p, the positively-charged residues are thought to unspecifically interact with the phosphate backbone of nucleic acids, including double-stranded DNA or RNA/DNA hybrids.

**DOT1A rapidly mono-methylates H3K76 and prefers H3K76me0 as a substrate**

Kinetic progression of individual H3K76 methylation species was monitored by quantitative MS for Δ42-DOT1A and full-length DOT1A (Fig. 2d and Supplementary Figs. 3 and 4). Reactions were performed with 0.125 μM enzyme and 1.0 μM of the recombinant *T. brucei*

**Fig. 2 | Methyltransferase activities, DNA binding, and mass spectrometry analysis of H3K76 methylation products of DOT1A and DOT1B. a** SDS-PAGE of recombinant full-length DOT1A, Δ42-DOT1A, and Δ27-DOT1B (see also Supplementary Fig. 1). **b** Methyltransferase activity of full-length DOT1A, Δ42-DOT1A, and Δ27-DOT1B toward the recombinant *T. brucei* nucleosome after 30 min (top) and human DOT1L (bottom) with DNA-binding region (residues 1–420) and without DNA-binding region (residues 1–351) toward recombinant human un-ubiquitinated nucleosome measured by [3]H incorporation (decays per minute [dpm]). Data are presented as mean values ± SD of *n* independent experiments (*n* = 4 for FL-DOT1A and Δ27-DOT1B, *n* = 3 for DOT1L (1–420), *n* = 2 for DOT1L (1–351), *n* = 19 for Δ42-DOT1A, *n* = 8 for no enzyme). Data were analyzed by one-way ANOVA followed by Tukey's post-hoc test. Adjusted *P* value between FL-DOT1A vs. Δ42-DOT1A is 0.001, Δ42-DOT1A vs. Δ27-DOT1B is 0.93, Δ42-DOT1A vs. no enzyme is <0.001, Δ27-DOT1B vs. no enzyme is <0.001, DOT1L (1–420) vs. DOT1L (1–351) is <0.001, DOT1L (1–420) vs. no enzyme is <0.001, and DOT1L (1–351) vs. no enzyme is 0.89. **c** Fluorescence polarization assay using full-length DOT1A (top, blue) and Δ42-DOT1A (top, magenta) and DOT1L (bottom) with DNA-binding region (residues 1–420, blue) and without DNA-binding region (residues 1–351, magenta). The $K_D$ values were calculated using a one-site model and are presented as mean values ± SEM of *n* independent experiments (*n* = 2 for DOT1L (1–351) and (1–420), and *n* = 1 for Δ42-DOT1A and FL-DOT1A). **d, e** Quantitative MS analysis of H3K76 methylation products generated by Δ42-DOT1A (**d**) and Δ27-DOT1B (**e**) expressed as the percentage of the sum of intensities of all related peaks plotted over the time course of the reaction, $k_{obs}$ values are presented as best-fit values ± SEM reported in GraphPad Prism (see also Supplementary Figs. 3 and 4). **f, g** Quantitative MS analysis of H3K76 methylation products generated by a combination of Δ42-DOT1A and Δ27-DOT1B either by adding Δ27-DOT1B after 30 min of a Δ42-DOT1A reaction (**f**) or by adding them simultaneously to the nucleosome substrate (**g**), $k_{obs}$ values are presented as best-fit values ± SEM (see also Supplementary Figs. 3 and 4). Source data are provided as a Source Data file.

canonical nucleosome as the substrate. The observed H3K76me0 and me1 disappearance rate constants $k_{obs}$ me0 and $k_{obs}$ me1 and the observed H3K76me2 appearance rate constant $k_{obs}$ me2 were estimated by fitting an exponential decay model to the data[40,41]. Since the kinetics of Δ42-DOT1A and full-length DOT1A behave very similarly (Fig. 2d and Supplementary Fig. 4), the impact of the DNA-binding region of DOT1A (residues 1–42) is minor. In Δ42-DOT1A-catalyzed reactions, unmodified substrate (H3K76me0) rapidly disappears ($k_{obs}$ me0 disappearance of 10.1 ± 0.6 min⁻¹). H3K76me1 immediately accumulates to ~85% within 5 min and slowly disappears ($k_{obs}$ me1 disappearance of 0.17 ± 0.08 min⁻¹). Concomitantly, H3K76me2 slowly appears over 2 h ($k_{obs}$ me2 appearance of 0.25 ± 0.04 min⁻¹), while no me3 is detected (Fig. 2d). Thus, the conversion from H3K76me1 to me2 is ~60 times slower than the H3K76me0 to me1 conversion, demonstrating that Δ42-DOT1A prefers me0 as a substrate and that methylation from me1 to me2 is the rate-limiting step for DOT1A. These MS-based data confirm our result from the radioactive methyltransferase assay that the DOT1A DNA-binding region (residues 1–42) is not required for robust H3K76 methyltransferase activity (Fig. 2b).

### DOT1B disfavors H3K76me0 as a substrate and cooperates with DOT1A to generate H3K76me3

As Δ27-DOT1B showed significant methyltransferase activity by measuring tritium incorporation from [methyl-³H] AdoMet in vitro (Fig. 2b), the Δ27-DOT1B kinetic progression was monitored under the same conditions as full-length DOT1A and Δ42-DOT1A. Strikingly, the kinetics of Δ27-DOT1B is markedly different from that of Δ42-DOT1A, as unmodified substrate (H3K76me0) disappears slowly ($k_{obs}$ me0 disappearance of 0.80 ± 0.08 min⁻¹) and H3K76me3 appears with a $k_{obs}$ me3 appearance of 0.6 ± 0.2 min⁻¹ at a similar rate of me0 disappearance, without large accumulation of me1 and me2 (Fig. 2e). These data demonstrate that Δ27-DOT1B is an active H3K76 trimethyltransferase and generates me3 without ubiquitin interaction. As the DOT1B intermediate products H3K76me1 and me2 do not accumulate, DOT1B likely prefers H3K76me1 and me2 as substrates over me0. To test this hypothesis, we monitored Δ27-DOT1B activity on a Δ42-DOT1A-methylated nucleosome sample as the starting substrate due to current limitations in synthesizing H3K76me1- and H3K76me2-modified nucleosomes (Fig. 2f). To this end, 1.0 μM of unmodified recombinant nucleosome was first reacted with 0.125 μM of Δ42-DOT1A for 30 min. At 30 min, prior to adding Δ27-DOT1B, the relative H3K76 modification fractions were ~2% of me0, ~75% of me1, ~20% of me2, and 0% of me3. Then, 0.125 μM of Δ27-DOT1B was added, and the methylation kinetics was monitored from 30 to 60 min. H3K76me1 disappeared within 5 min with a $k_{obs}$ me1 disappearance of 18.8 ± 0.0 min⁻¹, H3K76me2 surged and was converted to H3K76me3 within 15 min with a $k_{obs}$ me2 disappearance of 3.7 ± 0.3 min⁻¹ and $k_{obs}$ me3 appearance of 4.9 ± 0.4 min⁻¹. Under these conditions, the $k_{obs}$ me3 appearance is about eight times faster than with Δ27-DOT1B alone

($k_{obs}$ me3 appearance of 0.6 ± 0.2 min⁻¹) (Fig. 2e, f). Cooperativity between Δ42-DOT1A and Δ27-DOTB was also observed, when equal amounts of these enzymes were simultaneously reacted with unmodified nucleosome substrate: H3K76me3 is generated with a $k_{obs}$ me3 appearance of 1.44 ± 0.03 min⁻¹, ~2.5 times faster than with Δ27-DOT1B alone (Fig. 2e, g). Under these conditions, H3K76me1 and H3K76me2 peaked at ~10% and ~30% after 10 min, respectively, and disappeared within 1 h (Fig. 2g). These data support a model where Δ27-DOT1B methylation from me0 to me1 is the rate-limiting step, as Δ27-DOT1B prefers me1 and me2 over me0 as a substrate and demonstrate that Δ42-DOT1A and Δ27-DOT1B can more efficiently generate H3K76me3 together than Δ27-DOT1B alone.

### Crystal structures of DOT1A

Our findings that DOT1A and DOT1B efficiently methylate H3K76 without DNA binding and ubiquitin interaction[10,14] (Fig. 2) indicate that kinetoplastid DOT1 enzymes have a substrate recognition mechanism distinct from that in other eukaryotes such as yeast and humans. Significant sequence divergence is found in the N-terminal domain, key residues in the methyltransferase motifs IV, VI, and VIII are different, and conserved motif X residues are completely lacking (Fig. 1). To investigate the implications of these differences on the structural level, we determined the X-ray crystal structure of DOT1A and compared it to human and yeast active-form DOT1-nucleosome complexes[9,10]. Δ42-DOT1A crystallized in complex with the methyl donor product AdoHcy in two different crystal forms (space groups $C222_1$ and $P2_12_12_1$), and their structures were determined at 2.1 and 1.9 Å resolution, respectively (Fig. 3b and Supplementary Table 1). The asymmetric units of the $C222_1$ and $P2_12_12_1$ space groups contain one and two DOT1A enzymes referred to as Molecules I to III, respectively (Supplementary Figs. 5a and 6a). All three molecules are highly similar with a root-mean-square deviation (rmsd) of ~0.3 Å comparing 234 pairs of Cα atoms. Hence, we will describe the structure of Molecule I in the $C222_1$ space group and only discuss significant discrepancies among the three molecules.

### The DOT1A N-terminal domain folds into a β-hairpin structure

The N-terminal domain sequence of DOT1A (Glu46-Leu93) and DOT1B (Glu29-Leu76) are highly conserved among kinetoplastids (43% identity and 61% similarity)[18], but completely distinct from the N-terminal domain sequences of human and yeast DOT1 enzymes (Figs. 1b and 3a), which is reflected by vastly different N-terminal domain structures (Supplementary Fig. 7). The DOT1A N-terminal domain folds into a Zn-coordinating β-hairpin structure comprising residues Glu46 to Cys79 (Fig. 3b). The conserved Gly48, Gly50, Pro52, Pro55, Pro59, Pro64, and Gly68 residues are integral to the β-hairpin by forming sharp turns (Fig. 3c). The β-hairpin structure is tightly associated with the methyltransferase core domain through a hydrogen bond network by extending the antiparallel β-sheet of the methyltransferase core domain (Fig. 3d). The two Oε atoms of the Glu46 carboxyl group form

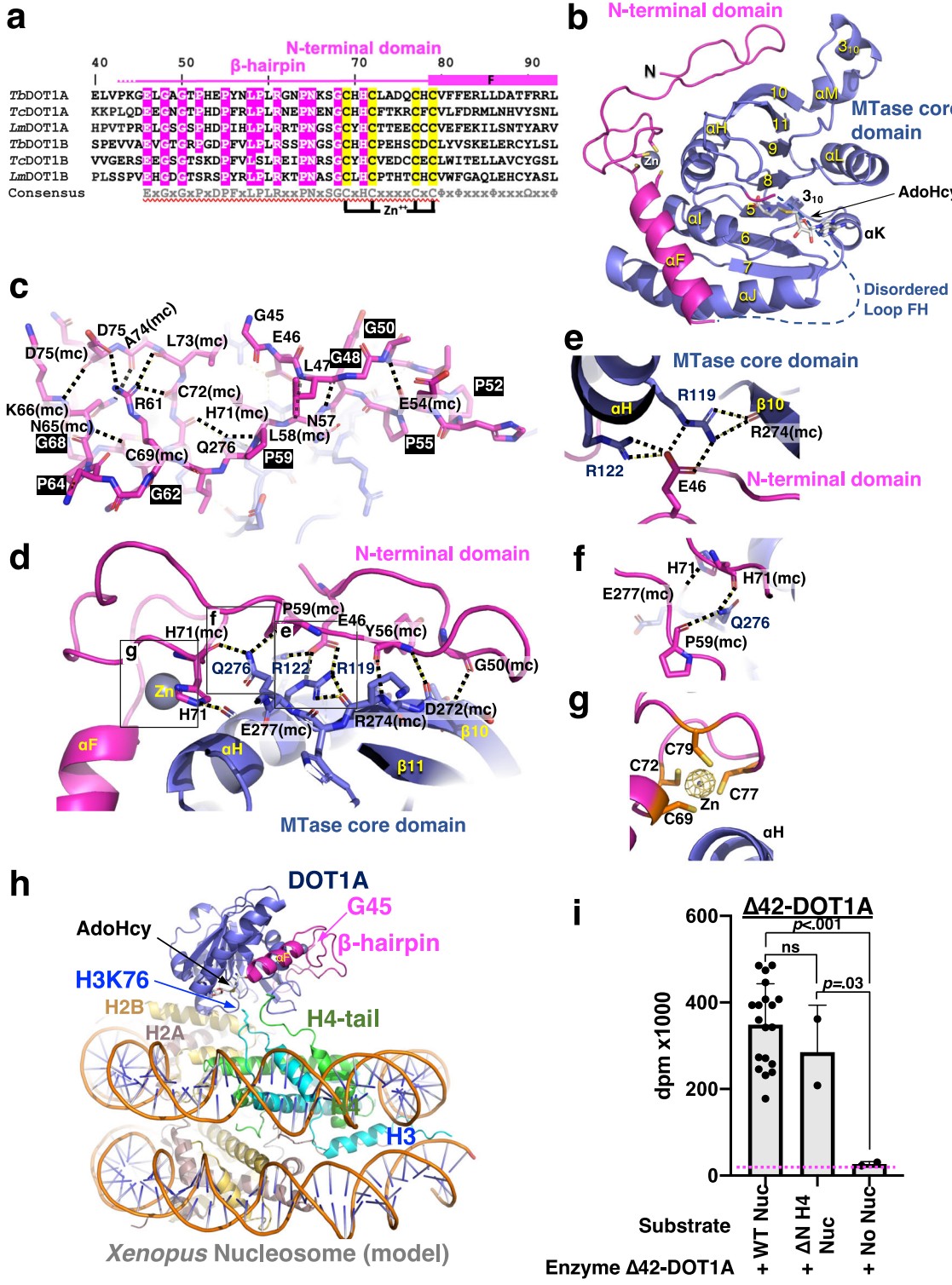

**Fig. 3 | Crystal structure of DOT1A. a** Sequence alignment of the kinetoplastid N-terminal domain among *T. brucei, T. cruzi,* and *L. major* DOT1A and DOT1B. **b** Crystal structure of Δ42-DOT1A. **c** Network of internal polar interactions within the N-terminal domain indicated by dashed lines. mc refers to main chain. **d** Interactions between the N-terminal domain and the MTase core domain indicated by dashed lines. **e** Network of internal hydrogen bonds centered on residues Arg119, Arg122, and Glu46. **f** Network of internal hydrogen bonds centered on residues Gln276, Glu277, and His71. **g** Zn-coordinating cysteine residues. A simulated annealing omit electron density map contoured at 10σ above the mean is shown for Zn²⁺ (yellow mesh, see also Supplementary Fig. 8). **h** Model of a DOT1A-nucleosome complex obtained by superimposing the DOT1A crystal structure

(Molecule I, PDB: 8FJN) onto the human active-state DOT1L-*Xenopus* nucleosome cryo-EM structure (PDB: 6NQA). **i** MTase activities of wild-type Δ42-DOT1A toward the wild-type *T. brucei* nucleosome (WT Nuc), histone H4 tail-less *T. brucei* nucleosome (ΔN H4 Nuc), and no nucleosome (No Nuc). Data are presented as mean values ± SD of *n* independent experiments (*n* = 19 for Δ42-DOT1A/WT Nuc, *n* = 2 for Δ42-DOT1A/ΔN H4 Nuc and Δ42-DOT1A/No Nuc). Data were analyzed by one-way ANOVA followed by Tukey's post-hoc test. Adjusted *P* value between Δ42-DOT1A/WT Nuc vs. Δ42-DOT1A/ΔN H4 Nuc is 0.63, Δ42-DOT1A/WT Nuc vs. Δ42-DOT1A/No Nuc is <0.001, Δ42-DOT1A/ΔN H4 Nuc vs. Δ42-DOT1A/No Nuc is 0.03. Source data are provided as a Source Data file.

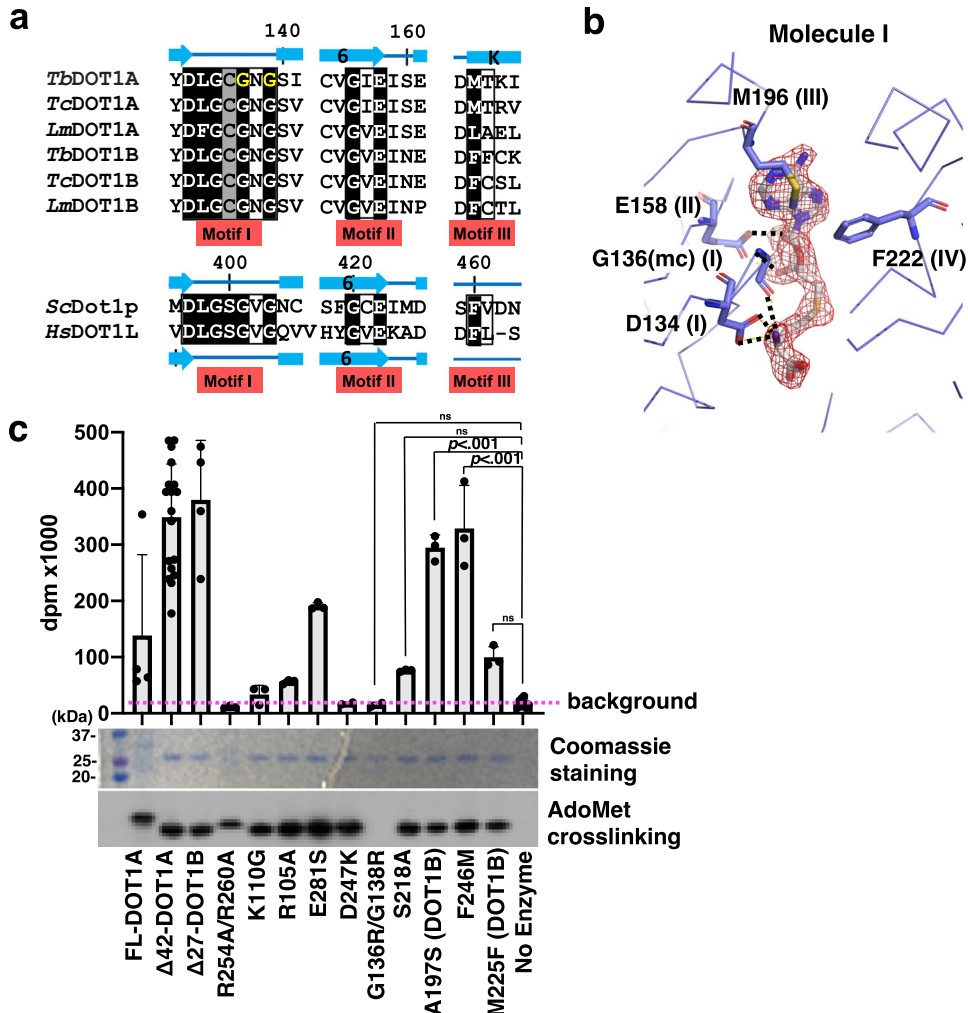

**Fig. 4 | AdoMet binding. a** Sequence conservation of methyltransferase motifs I, II, and III among *T. brucei, T. cruzi, L. major* DOT1A and DOT1B, yeast Dot1p, and human DOT1L. **b** Simulated-annealing omit map of AdoHcy contoured at 4σ above the mean (red mesh) and AdoHcy-interacting residues (methyltransferase motifs are indicated in parenthesis). **c** Methyltransferase activity of wild-type DOT1A and DOT1A/B mutants toward the wild-type *T. brucei* nucleosome measured by ³H incorporation (decays per minute [dpm]), top). AdoMet binding analyzed by cross-linking experiments showing Coomassie stain (middle) and fluorography (bottom). Δ27-DOT1B mutants are labelled with "(DOT1B)", while the remaining mutants refer to Δ42-DOT1A mutants. Data are presented as mean values ± SD of n independent experiments (n = 4 for FL-DOT1A and Δ27-DOT1B, n = 19 for Δ42-DOT1A, n = 3 for Δ42-DOT1A R254A/R260A, Δ42-DOT1A K110G, Δ42-DOT1A R105A, Δ42-DOT1A E281S, Δ42-DOT1A S218A, Δ27-DOT1B A197S, Δ42-DOT1A F246M, Δ27-DOT1B M225F, n = 2 for Δ42-DOT1A D247K, Δ42-DOT1A G136R/G138R, n = 8 for No Enzyme). Data were analyzed by one-way ANOVA followed by Tukey's post-hoc test. Adjusted *p*-value between Δ42-DOT1A G136R/G138R vs. No Enzyme is >0.99, Δ42-DOT1A S218A vs. No Enzyme is <0.001, Δ27-DOT1B A197S vs. No Enzyme is >0.99, Δ42-DOT1A F246M vs. No Enzyme is <0.001, Δ42-DOT1A M246F vs. No Enzyme is 0.93. Source data are provided as a Source Data file.

hydrogen bonds with both the nitrogen Nη₂ and Nε atoms of Arg119 and the nitrogen Nη₁ and Nη₂ atoms of Arg122 in the αH helix (Fig. 3e). Three hydrogen bonds from Pro59 and His71 to Gln276 and Glu277 further stabilize the β-hairpin-methyltransferase core domain interaction (Fig. 3f). The four cysteine residues Cys69, Cys72, Cys77, and Cys79 coordinate zinc, connecting the β-hairpin and helix αF (Fig. 3g and Supplementary Fig. 8). These four Zn-coordinating cysteine residues are essential for protein stability, as a Δ42-DOT1A mutant containing Ser residues in place of all four Zn-coordinating Cys was insoluble. Therefore, the N-terminal β-hairpin structure appears to stabilize the DOT1A methyltransferase core domain.

**The histone H4 N-terminal tail is dispensable for DOT1A activity**
A model of the DOT1A-nucleosome complex, generated by superimposing the DOT1A structure onto the active form of the human DOT1L-nucleosome complex (PDB: 6NQA)[9], indicates that the β-hairpin structure is not in vicinity to the putative nucleosome-DOT1 interface (Fig. 3h). Thus, the N-terminal domain of DOT1A is likely to play a

different role from that of the N-terminal helical domains of yeast Dot1p and human DOT1L, which directly interact with the histone H4 N-terminal tail and which are essential for activity[9–11,42]. In fact, the histone H4 N-terminal tail is dispensable for DOT1A and DOT1B activities in vivo[37] and in vitro (Fig. 3i). These data further illustrate that DOT1A and DOT1B use a distinct substrate nucleosome recognition mechanism.

**AdoMet recognition mediated by methyltransferase motifs I, II, and III is conserved in DOT1A**
The methyltransferase motifs I, II, and III are highly conserved among DOT1 enzymes including DOT1A and DOT1B (Fig. 4a), and electron density of AdoHcy in the crystal structures of DOT1A are clearly observed in all three molecules (Fig. 4b and Supplementary Figs. 5b and 6b). Specifically, AdoHcy is recognized by DOT1A Asp134 (motif I), the Gly136 main chain (motif I), Glu158 (motif II), Met196 (motif III), and Phe222 (adjacent to motif IV) (Fig. 4b). A motif-I Gly136-to-Arg/Gly138-to-Arg Δ42-DOT1A double mutant lost AdoMet binding capacity and methyltransferase activity (Fig. 4c).

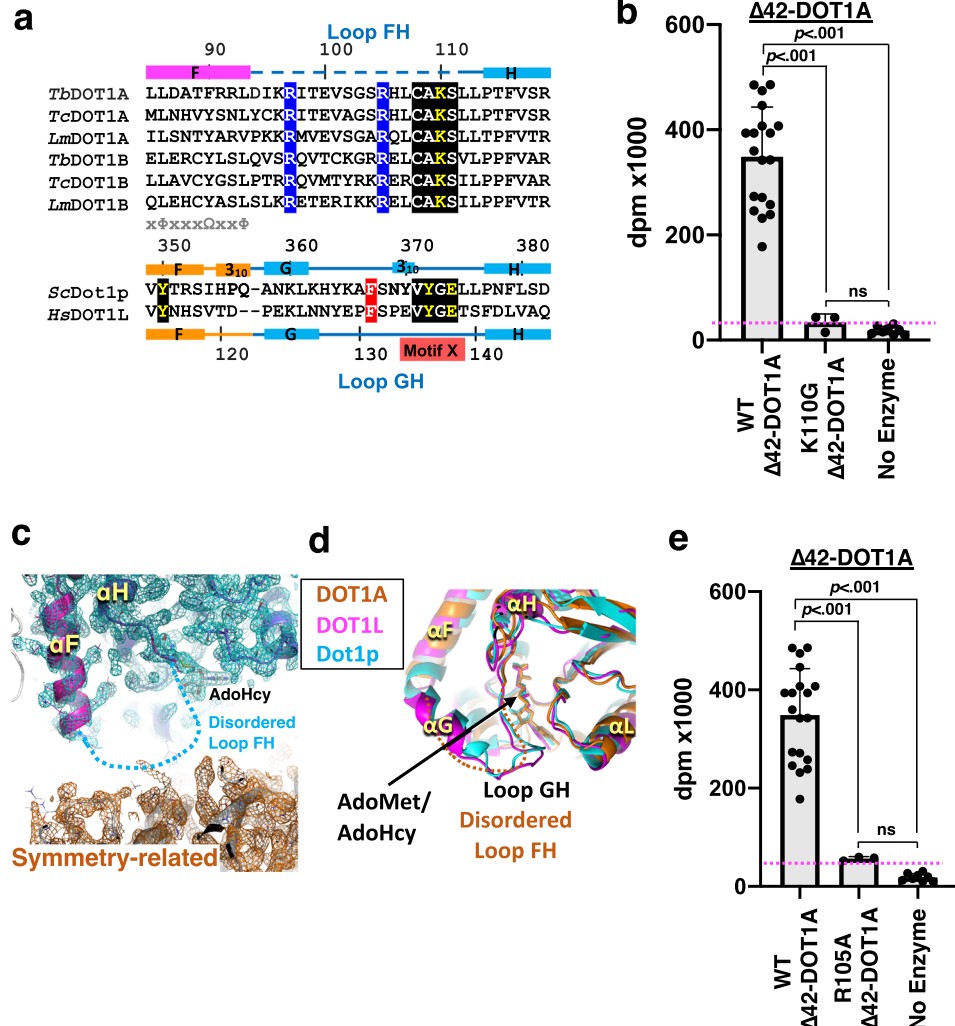

**Fig. 5 | Key residues in the CAKS sequence-containing loop FH for DOT1A methyltransferase activity. a** Sequence conservation of loop FH sequences among *T. brucei, T. cruzi, L. major* DOT1A and DOT1B, yeast Dot1p, and human DOT1L. In kinetoplastid DOT1A and DOT1B, the two conserved Arg residues (blue background) and the conserved CAKS sequence (black background) are highlighted. In yeast Dot1p and human DOT1L, motif X (black background), the key phenylalanine in loop GH (red background), and the conserved tyrosine in helix αF (yellow on black background) are highlighted. **b** Methyltransferase activity of wild-type Δ42-DOT1A and mutant Δ42-DOT1A harboring the K110G mutation within the CAKS sequence toward the wild-type *T. brucei* nucleosome measured by [3]H incorporation (decays per minute [dpm]). Data are presented as mean values ± SD of *n* independent experiments (*n* = 19 for WT Δ42-DOT1A, *n* = 2 for K110G Δ42-DOT1A, *n* = 8 for No Enzyme). Data were analyzed by one-way ANOVA followed by Tukey's

post-hoc test. Adjusted *P* value between WT Δ42-DOT1A vs. K110G Δ42-DOT1A is <0.001, WT Δ42-DOT1A vs. No Enzyme is <0.001, K110G Δ42-DOT1A vs. No Enzyme is 0.96. **c** The loop FH is disordered as shown by a 2*Fo-Fc* electron density map comprising Molecule I (blue mesh) and a symmetry-related Molecule I (orange mesh) contoured at 1σ above the mean (see also Supplementary Figs. 5 and 6). **d** DOT1A region featuring the disordered loop FH between αF and αH helices, superimposed onto DOT1L and Dot1p. **e** Methyltransferase activitiy of wild-type Δ42-DOT1A and the R105A Δ42-DOT1A mutant. Data are presented as mean values ± SD of *n* independent experiments (*n* = 19 for Δ42-DOT1A, *n* = 3 for Δ42-DOT1A R105A, *n* = 8 for No Enzyme). Data were analyzed by one-way ANOVA followed by Tukey's post-hoc test. Adjusted *P* value between WT Δ42-DOT1A vs. R105A Δ42-DOT1A is <0.001, WT Δ42-DOT1A vs. no enzyme is <0.001, R105A Δ42-DOT1A vs. No Enzyme is 0.75. Source data are provided as a Source Data file.

## Motif X residues of the activation loop are substituted by a kinetoplastid-specific sequence essential for methyltransferase activity

In human DOT1L and yeast Dot1p, AdoMet/AdoHcy is covered by a rigid lid consisting of the motif X (Val-Tyr-Gly-Glu: VYGE)-containing loop GH (also termed activation loop in DOT1L, residues 122–140) that allows for AdoMet/AdoHcy co-purification with the protein[7,8,43]. Motif X plays a key role in establishing the rigid lid structure for H3K79 lysine channel formation (Supplementary Fig. 9). A Glu138-to-Ala mutation within the DOT1L motif X and Glu374-to-Ala/Gln mutations within the Dot1p motif X abolish their activities while still retaining AdoMet-binding capacity[8,44]. A conserved phenylalanine residue within the loop plays a key role in projecting the target H3K79 lysine toward AdoMet

by penetrating its phenyl group between the histone H3 L1 and histone H4 L2 loops (Supplementary Fig. 10)[9,10]. Indeed, a DOT1L Phe131-to-Ala mutation abolished methyltransferase activity[9]. However, DOT1A and DOT1B lack the motif X sequence (VYGE) in the corresponding loop FH (residues 94–111 in *T. brucei* DOT1A), and a phenylalanine is not present (Fig. 5a). In kinetoplastids, the motif X-corresponding sequence is invariably substituted by the Cys-Ala-Lys-Ser (CAKS) sequence (Fig. 5a). Akin to motif X being essential for human and yeast DOT1 enzymes, we found that the CAKS sequence in kinetoplastids is also essential for DOT1A activity, as demonstrated by a Lys110-to-Gly mutation within the DOT1A CAKS sequence that abolishes *Δ42*-DOT1A methyltransferase activity (Fig. 5b) while still retaining AdoMet-binding capacity (Fig. 4c).

## DOT1A harbors a flexible loop FH

In DOT1A molecules I and II of the two crystal forms, the CAKS sequence-containing loop FH lacks electron density and is disordered, implying flexibility (Fig. 5c and Supplementary Fig. 6a). In DOT1A molecule III, this loop is ordered (colored in purple), which is likely due to stabilization by a symmetry-related molecule in the crystal (Supplementary Fig. 6a). By contrast, loop GH in DOT1L and Dot1p forms a rigid structure (Fig. 5a, d). Apart from the CAKS sequence, the loop FH sequence is not highly conserved across kinetoplastid DOT1A and DOT1B, with only two arginine residues, Arg97 and Arg105 in *Tb*DOT1A, being conserved (Fig. 5a). Arg97 forms a salt bridge with Glu281 within motif VIII (VEWC) in molecule III, but it is not essential as evidenced by a Glu281-to-Ser mutant that does not abrogate methyltransferase activity (Supplementary Fig. 11). In contrast, Arg105 is critical for methyltransferase activity, as a DOT1A Arg105-to-Ala mutant loses activity, despite retaining its capacity for AdoMet binding (Figs. 4c and 5e). These data further suggest that kinetoplastid DOT1 enzymes use a substrate recognition mechanism distinct from yeast and human DOT1 enzymes[7,9].

## Identification of DOT1A-nucleosome interacting key residues

For human DOT1L and yeast Dot1p, the following interactions between enzyme and substrate are essential: (1) DNA interactions with the DNA-binding region, which is not resolved in the electron density of any DOT1L/Dot1p structure, (2) the histone H4 N-terminal tail interaction with the DOT1 acidic cleft between the N-terminal helical domain and the C-terminal methyltransferase core domain, which includes the critical interaction between H4R19 and the histone H3 L1 loop main-chain that facilitates projecting the histone H3K79 side chain toward the methyl donor AdoMet[9–11], (3) the nucleosome acidic patch interaction with an arginine anchor (DOT1L Arg278/Arg282 residues), (4) and the H2B mono-ubiquitin interaction with the DOT1 C-terminal helix for di- or tri-methylation (Fig. 6a)[9,10]. Since kinetoplastid DOT1A and DOT1B neither require DNA binding (Fig. 2), nor histone H4 N-terminal tail recognition in vitro (Fig. 3i) and in vivo[37], nor ubiquitin interaction for their methyltransferase activities (Fig. 2), insight into the DOT1A-nucleosome substrate interface would be important to decipher the distinct mechanism of DOT1A/DOT1B substrate recognition. We used X-ray footprinting to identify nucleosome residues protected from radiation damage via DOT1A binding. The nucleosome used for X-ray footprinting had H3K76 replaced by norleucine (Nle) to increase complex stability[9,45]. As expected, Δ42-DOT1A protects H3 L1 loop residues, in particular Nle76 and its adjacent residue Glu77, the most (Fig. 6c and Supplementary Fig. 12). Other highly protected residues include Asp93 of histone H2B, which is part of the acidic patch, and H2B Lys96 nearby (Fig. 6b).

## Acidic patch-arginine anchor interactions are conserved across DOT1 enzymes

Based on our DOT1A-nucleosome model (Fig. 6b), H2B Asp93 would interact with the potential arginine anchor residues Arg254 and Arg260 of DOT1A conserved across kinetoplastid DOT1 enzymes. The importance of these two arginine anchor residues was confirmed by a R254A/R260A double mutant that abolishes nucleosome binding revealed by X-ray footprinting and methyltransferase activity while still retaining AdoMet binding (Figs. 4c and 6d, e and Supplementary Fig. 13). These data suggest that acidic patch-arginine anchor interactions are used for substrate recognition across all DOT1 enzymes.

## A motif VI acid/base substitution is essential for stabilizing the DOT1A-nucleosome complex

Notably, the basic lysine residue within motif VI (Gx$_6$K) of human and yeast DOT1 enzymes is substituted by an acidic residue in kinetoplastid homologs, Asp247 in *T. brucei* DOT1A and Glu225 in *T. brucei* DOT1B (Fig. 6d). Structural analysis of the human DOT1L ubiquitinated

*Xenopus* nucleosome active and poised forms in fact suggest an important role of the motif VI lysine in H3K79 methylation (Fig. 6a)[9,44,46–48]. In the active-form complex structures, which could only be captured when H3K79 was mutated to an inert norleucine or methionine to increase enzyme-substrate interactions, the canonical motif VI lysine is close to the conserved histone H2B Lys108. In poised-state DOT1-ubiquitinated nucleosome complex structures, which contain the natural substrate residue H3K79, the canonical motif VI lysine is further away from H2B Lys108. These observations suggest that the two positively-charged lysines opposing each other at the DOT1-nucleosome interface would repel each other (Fig. 6a), weakening DOT1-nucleosome active-form interactions and favoring a poised state of the DOT1-ubiquitinated nucleosome complex[9,44,46–48]. This Lys-Lys repulsion may also accelerate $k_{cat}$ and AdoMet/AdoHcy exchange[7,9]. When we converted the positive charge of DOT1L Lys270 into a negative charge in the Lys270-to-Asp DOT1L mutant, methyltransferase activity was reduced, potentially due to a decrease in the DOT1L K270D-nucleosome turnover ratio (Fig. 6f).

As the H2B Lys108 of yeast and humans is conserved in kinetoplastids (Lys96 in *T. brucei*, Fig. 6g), the Lys-to-Asp/Glu substitution in the non-canonical motif VI (Gx$_6$D/E) of kinetoplastids is expected to stabilize the substrate-enzyme complex. Indeed, X-ray footprinting verified that *T. brucei* histone H2B Lys96 represents one of the most protected residues by DOT1A (Fig. 6c and Supplementary Fig. 12). When mutating the acidic to a basic residue by an Asp247-to-Lys mutation in DOT1A, which reverts the non-canonical to the canonical motif VI sequence, DOT1A-nucleosome binding probed by X-ray footprinting and methyltransferase activity are abolished (Fig. 6h and Supplementary Fig. 13). While the non-canonical DOT1A/B motif VI acidic residue would facilitate DOT1A-nucleosome association in the absence of ubiquitin interaction and nanomolar-affinity DNA binding, this interaction cannot be too tight in order to allow for substrate release and co-factor exchange for subsequent methylation steps. Thus, motif VI-nucleosome interactions seem fine-tuned in different systems: attractive in ubiquitination-independent DOT1 enzymes (e.g., kinetoplastids) vs. repulsive in ubiquitin-dependent DOT1 enzymes (e.g., yeast and human).

## DOT1A Ser218/DOT1B Ala197 within motif IV impact substrate preference but not product specificity

Since active-site pocket size has been shown to regulate substrate and product specificity or preference in some methyltransferases[49–51], Ser218/Ala197 within the motif IV and Phe246/Met225 within the motif VI of DOT1A/DOT1B were proposed as key residues for product specificity based on an antibody-based assay (Fig. 7a, b)[19]. The quantitative MS approach employed here allows us to revisit this conclusion. We found that the DOT1A F246M and DOT1B M225F mutants did not have an impact on substrate preference or product specificities, consistent with DOT1A Phe246 not directly contributing to the active-site surface (Fig. 7b–d). DOT1A F246M prefers me0 and does not generate me3, while DOT1B M225F generates me3 without me1 and me2 accumulation, akin to the wild-type enzymes (Fig. 2d, e). While product specificity has not changed in the DOT1A S218A mutant, it has markedly altered substrate preference, as activity toward me0 is significantly reduced and a higher concentration of DOT1A S218A mutant is required for me0 to me1 conversion, suggesting a significant increase in the apparent $K_M^{H3K76me0}$ of the DOT1A S218A mutant (Fig. 7e and Supplementary Fig. 14). Even though DOT1A S218A at the highest concentration (1.0 μM) completely converts me0 to me2 within ~30 min, no me3 is detected after 1 h (Supplementary Fig. 14c). Similarly, the DOT1B A197S mutant alters substrate preference, as me0 is rapidly converted to me1 within ~10 min, but does not alter product specificity, as me3 is still produced (Fig. 7f). The rapidly produced me1 is slowly converted to me2, with me3 gradually increasing over time. These results demonstrate that DOT1A S218 and DOT1B A197 within

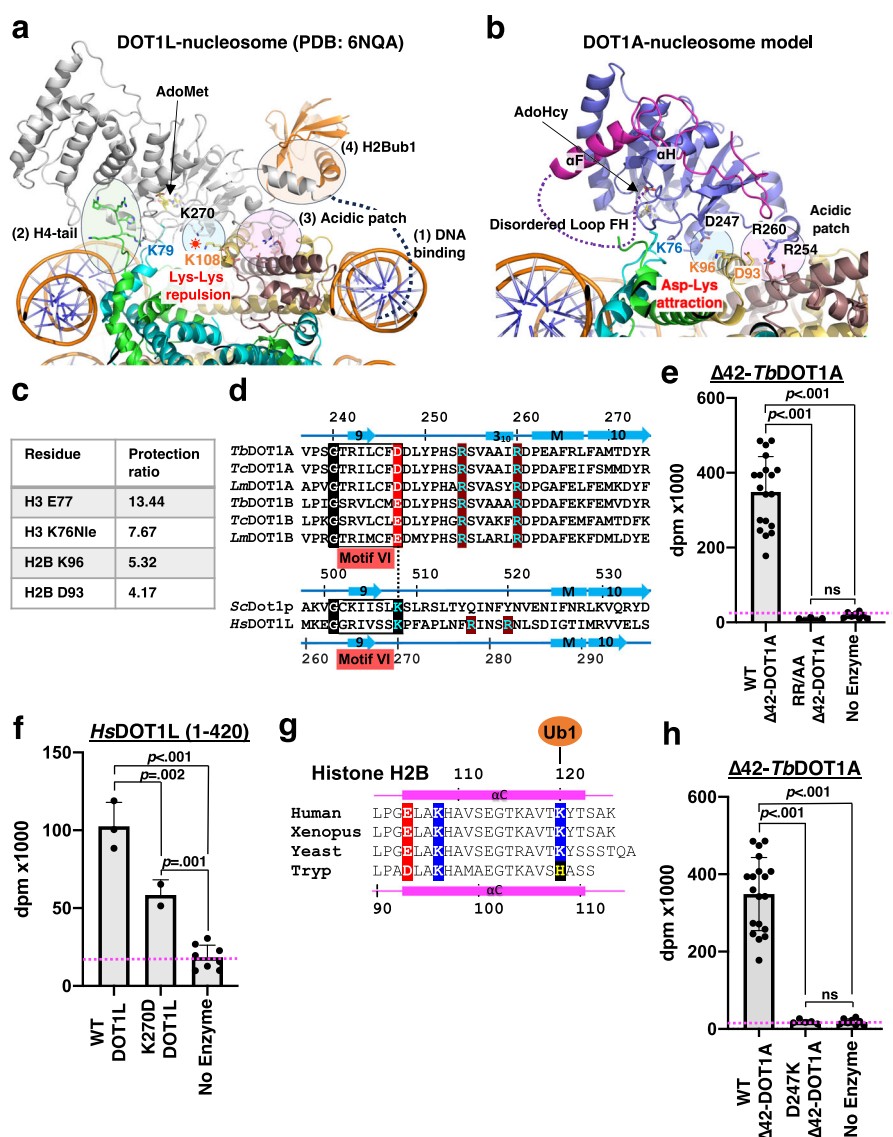

**Fig. 6 | Distinct electrostatic interactions between motif VI residues and histone H2B Lys. a** Human active-state DOT1L-*Xenopus* nucleosome cryo-EM structure (PDB: 6NQA). Key interactions are shown. 1: DNA binding, 2: H4-tail-DOT1L interactions, 3: acidic patch-DOT1L arginine anchor interactions, and 4: H2Bub1-DOT1L interactions. The positively-charged H2B K108 and DOT1L K270 residues would cause a repulsion. **b** Model of a DOT1A-nucleosome complex obtained by superimposing the DOT1A crystal structure (Molecule I, PDB: 8FJN) onto the human active-state DOT1L-*Xenopus* nucleosome cryo-EM structure (PDB: 6NQA). **c** Most protected histone residues by wild-type DOT1A from X-ray footprinting. **d** Sequence alignment of motif VI, two potential DOT1A/DOT1B arginine anchor residues, and two DOT1L arginine anchor residues. **e** Methyltransferase activity of wild-type Δ42-DOT1A and the Δ42-DOT1A R254A/R260A double mutant (RR/AA) toward the wild-type *T. brucei* nucleosome. Data are presented as mean values ± SD of n independent experiments ($n = 19$ for WT Δ42-DOT1A, $n = 3$ for R254A/R260A Δ42-DOT1A, $n = 8$ for No Enzyme). Data were analyzed by one-way ANOVA followed by Tukey's post-hoc test. Adjusted *P* value between WT Δ42-DOT1A vs. R254A/

R260A Δ42-DOT1A is <0.001, WT Δ42-DOT1A vs. No Enzyme is <0.001, R254A/R260A Δ42-DOT1A vs. No Enzyme is >0.99. **f** Methyltransferase activity of human wild-type DOT1L (residues 1–420) and human DOT1L (residues 1–420) Lys270Asp mutant toward human un-ubiquitinated nucleosome. Data are presented as mean values ± SD of *n* independent experiments ($n = 3$ for WT DOT1L (1–420), $n = 2$ for K270D DOT1L (1–420), $n = 8$ for No Enzyme). Data were analyzed by one-way ANOVA followed by Tukey's post-hoc test. Adjusted *P* value between WT DOT1L (1–420) vs. K270D DOT1L (1–420) is 0.002, DOT1L (1–420) vs. No Enzyme is <0.001, K270D DOT1L (1-420) vs. no enzyme is 0.001. **g** Sequence alignment of histone H2B αC helix of human, Xenopus, yeast, and trypanosomes (Tryp). **h** Methyltransferase activity of wild-type Δ42-DOT1A and D247K Δ42-DOT1A mutant. Data are presented as mean values ± SD of *n* independent experiments ($n = 19$ for WT Δ42-DOT1A, $n = 5$ for Δ42-DOT1A D247K, $n = 8$ for No Enzyme). Data was analyzed by one-way ANOVA followed by Tukey's post-hoc test. Adjusted *P* value between WT Δ42-DOT1A vs. D247K Δ42-DOT1A is <0.001, WT Δ42-DOT1A vs. No Enzyme is <0.001, D247K Δ42-DOT1A vs. No Enzyme is >0.99. Source data are provided as a Source Data file.

---

motif IV impact substrate preference but not product specificity, suggesting that two DOT1 enzymes with distinct substrate preference would enable efficient tri-methylation of H3K76.

Interestingly, the corresponding residues in yeast and human motif IV are both asparagine (Fig. 7a). While H3K79me3 generation is detected by yeast Dot1p, me3 generation by DOT1L was not confirmed by MS (Fig. 7g, h)[16,46,52–54]. Although H3K79me3 detection by the anti-H3K79me3 antibody ab2621 is broadly used, it

significantly cross-reacts with H3K79me2[55]. We found that the human DOT1L N241A motif IV mutant did not alter its product specificity, as both wild-type and N241A mutant DOT1L generate me2, but not me3 (Fig. 7h, i). While Ser218 of *T. brucei* DOT1A and Ala197 of *T. brucei* DOT1B did not alter product specificity, hydrophobicity and size at this position may select preferences for methylated or unmethylated substrates in *T. brucei* DOT1 enzymes.

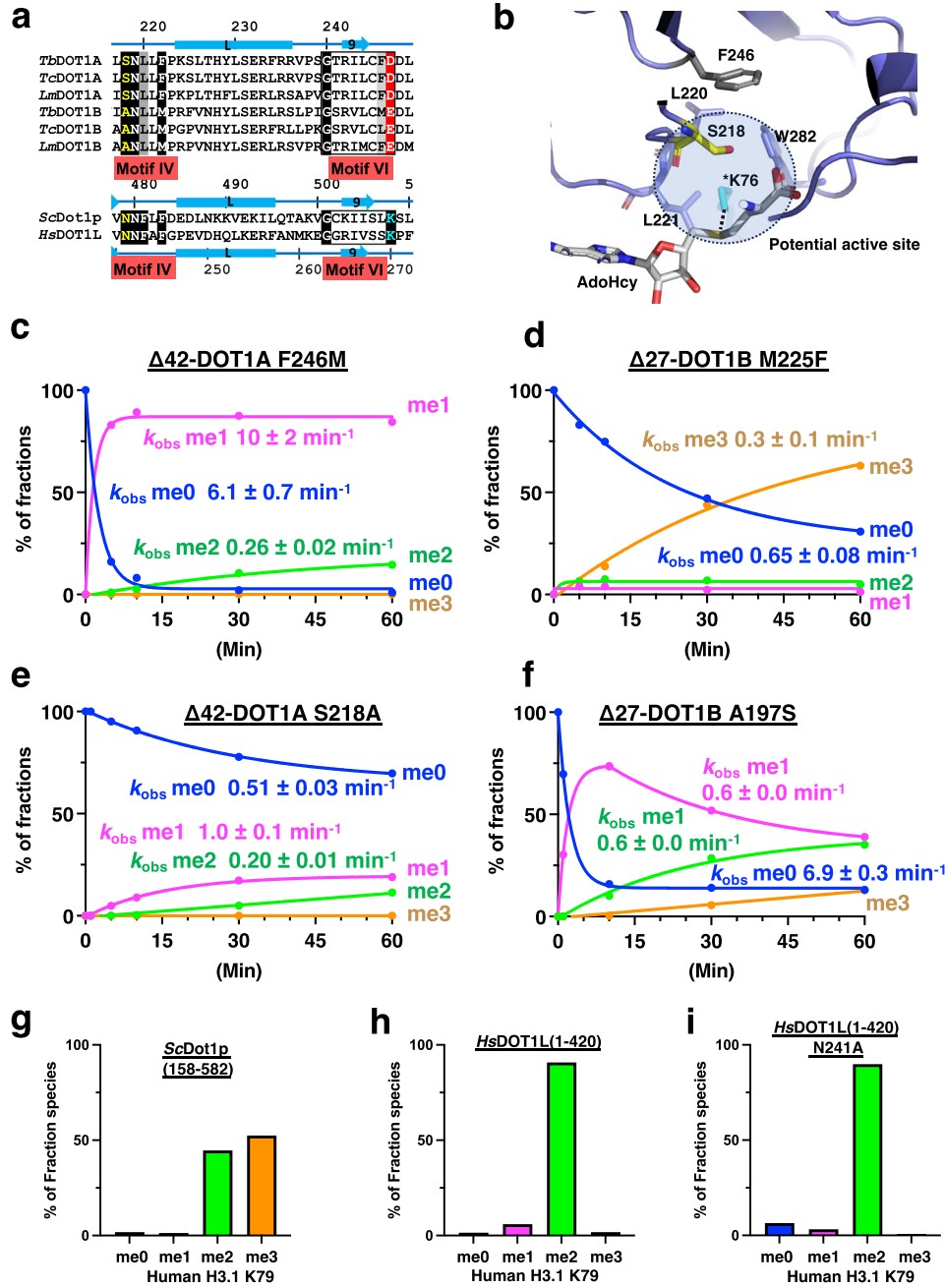

**Fig. 7 | A Ser/Ala switch within motif IV regulates DOT1A/B substrate preference. a** Sequence alignment of motifs IV and VI of *T. brucei, T. cruzi, L. major* DOT1A and DOT1B, yeast Dot1p, and human DOT1L. **b** Ser218 and Phe246 residues in the active site of DOT1A. K76* is modeled by superimposing the active-state DOT1L-nucleosome complex onto the DOT1A structure. **c–f** Quantitative MS analysis of H3K76 methylation products generated by Δ42-DOT1A F246M (**c**), Δ27-DOT1B M225F (**d**), Δ42-DOT1A S218A (**e**), and Δ27-DOT1B A197S (**f**) expressed as the percentage of the sum of intensities of all related peaks plotted over the time course of the reaction (see also Supplementary Fig. 3). **g–i** Quantitative MS analysis of H3.1 K79 methylation products generated by yeast Dot1p (158–582) (**g**), human DOT1L (1–420) (**h**), and the human DOT1L (1–420) N241A mutant (**i**) towards the ubiquitinated human nucleosome after 1 h reactions (see also Supplementary Fig. 3).

## Discussion

DOT1-catalyzed H3K79 methylation on the nucleosome disk surface is highly conserved among eukaryotes, yet its genomic temporal and spatial pattern in cells as well as the cellular processes linked to H3K79 methylation are very diverse in different organisms. Consistent with this diversity in H3K79 methylation, distinct systems to generate and regulate this modification have evolved. In humans and yeast, a single DOT1 enzyme (DOT1L and Dot1p, respectively) efficiently generates di- or tri-methylation at histone H3K79 enriched at active transcription start sites owing to H2B mono-ubiquitin interaction. In the early-branched kinetoplastid *T. brucei*, ubiquitin-independent H3K76 methylation is catalyzed by two enzymes (DOT1A and DOT1B), the majority of H3K76 is tri-methylated, and H3K76me3 is broadly distributed across the entire genome. Here, we have unveiled a mechanism for ubiquitin-independent nucleosome recognition by kinetoplastid DOT1 enzymes, discovered distinct H3K76 methylation kinetics of *T. brucei* DOT1A and DOT1B, and observed cooperative H3K76 methylation in vitro by these enzymes, which provides insights into their evolution, interplay, regulation, and biological function in the cellular context of *T. brucei*.

Based on phylogenetic studies, the two DOT1A and DOT1B methyltransferases in trypanosomes are thought to have originated through lateral transfer of a single DOT1 precursor from the animal lineage[56]. The unique N-terminal β-hairpin structure with the Zinc-coordinating cysteines are not present in human DOT1L and yeast Dot1p N-terminal regions and suggest that these features were likely acquired after this transfer. A precursor DOT1 enzyme would have lost the ubiquitin-binding helix due to the lack of H2B ubiquitination[18,21]. To achieve efficient H3K76 methylation in kinetoplastids in the absence of a ubiquitin-enzyme interaction, key residue substitutions in methyl-transferase motifs IV, VI, and X, which are conserved across kineto-plastid DOT1 methyltransferases, were required (Fig. 1b), although the methyltransferase motifs are widely conserved in DNA, RNA, and protein methyltransferases from bacteria to human[57,58]. One of such key substitutions refers to a Lys-to-Asp/Glu change in motif VI, which we have shown is essential for stabilizing the DOT1A-nucleosome complex with respect to yeast and human DOT1 enzymes (Fig. 6). The lack of H2B ubiquitination in kinetoplastids may have also triggered gene duplication, producing two distinct DOT1 enzymes with differing substrate preference and product specificity. Indeed, we discovered that a single residue in motif IV, Ser218 in DOT1A and Ala197 in DOT1B, determines substrate preference, but not product specificity in *T. brucei* (Fig. 7). The distinct substrate preferences can be rationalized by a hydrophilic, slightly larger residue (Ser) in the active site pre-ferring a more hydrophilic, unmethylated lysine and by a more hydrophobic, smaller residue (Ala) preferring a more hydrophobic, methylated lysine residue, because lysine methylation decreases its hydrogen-bonding capacity and increases its size and hydrophobicity[59]. Collectively, the kinetoplastid-specific residue sub-stitutions in DOT1A/B methyltransferases suggest distinct mechanisms of nucleosome substrate recognition in these enzymes and enable efficient H3K76 methylation in the absence of ubiquitin interaction, DNA binding with nanomolar affinity, and histone H4 tail interactions.

While the DOT1A/B-specific Ser/Ala residue of motif IV can ratio-nalize the distinct substrate preference of DOT1A and DOT1B, a mechanistic basis of their distinct product specificities (H3K76me2 vs. H3K76me3) remains elusive. To gain insight into their product speci-ficities, DOT1A/B-nucleosome complex structures would be useful. However, instability of these complexes makes cryo-EM structural studies challenging, and crystallization trials have not been successful yet. Even if these complex structures were available, it may still be difficult to understand how DOT1A and DOT1B determine their dif-ferent product specificity, as DOT1L-nucleosome and Dot1p-nucelosome structures could not explain why DOT1L only generates H3K79me2, but not H3K79me3 in vitro and in vivo[9,10,53]. We note that among the DOT1L-nucleosome and Dot1p-nucleosome structures, distinct conformations of the motif X-containing loop GH and the motif VIII-containing loop 10–11 in the active forms of DOT1L/Dot1p-nucleosome complexes feature significant differences that may account for the distinct DOT1L/Dot1p product specificity (Supple-mentary Fig. 15). Similarly, distinct conformations of the correspond-ing loops FH in *T. brucei* DOT1A and DOT1B, which considerably differ in their sequences, may impact product specificity.

In *T. brucei* cells, the abundance of the distinct H3K76 methylation products significantly varies during the cell cycle[36]. Immuno-fluorescence data show that H3K76me1 does not coincide with the proliferating cell nuclear antigen (PCNA), suggesting that newly incorporated H3 remains unmethylated at H3K76 during S phase. Fractions of H3K76me1 and me2 are not detected until G2 phase[60]. Specifically, H3K76me1 is detected in the early G2 phase, and H3K76me2 is detected in the late G2 phase, while H3K76me3 is con-stantly detected throughout the cell cycle. mRNA-seq-based data demonstrate that DOT1A and DOT1B mRNA is constantly detected throughout the cell cycle and are both slightly increased in G2, sug-gesting that DOT1A and/or DOT1B methyltransferase activities, but not

their expression, are cell cycle-regulated. Recently, RNase H2, which hydrolyzes RNA in DNA/RNA hybrids, was identified as a DOT1A and DOT1B binding partner[61,62]. RNase H2 interacts with PCNA and is mainly active during S phase to resolve R-loops for initiating transcription and regulating antigenic variation in *T. brucei*[63]. As the mRNA of RNase H2 subunits A, B, and C is increased in S phase but decreased in G2 phase, a DOT1A/B-RNase H2 interaction may suppress DOT1A/B methyltransferase activity[36]. Addressing how DOT1A/DOT1B and RNase H2 interact and how DOT1A/DOT1B-RNase H2 complex forma-tion impacts H3K76 methyltransferase activity and/or RNase H2 activity would advance our understanding of DOT1A function in cell cycle regulation.

Previous findings that the gene encoding the H3K76me2 methyl-transferase DOT1A is essential, that the gene encoding the H3K76me3 methyltransferase DOT1B is not essential, and that H3K76me2 accu-mulates in *DOT1B* knock-out cells[18] suggest that H3K76me1/me2 is critical for cell cycle progression. Given that DOT1B can generate H3K76me1/me2/me3 from me0 in vitro, it remained unclear why DOT1B would not be able to compensate for DOT1A in vivo. Our DOT1A and DOT1B kinetics data (Fig. 2d–f) can now resolve this mystery and offer specific scenarios to explain the in vivo data. Our analysis con-firmed that DOT1B can methylate H3K76 from me0 to me3, but importantly revealed that me1 and me2 intermediates do not accu-mulate (Fig. 2e). If H3K76me1/me2 is physiologically critical and if DOT1B can generate H3K76me3 from me0 in vivo, *DOT1A* knock-down cells would not accumulate H3K76me1/me2, leading to cell cycle defects, even if DOT1B was active. If in an alternative scenario DOT1B depends on H3K76me1/me2 generated by DOT1A due to its rate-limiting conversion of me0 to me1, DOT1B would be unable to effi-ciently methylate H3K76 in *DOT1A* knock-down cells. Indeed, our in vitro data show that combination of DOT1A and DOT1B−con-comitantly or sequentially−yields significant amounts of H3K76me1 and me2 and accelerates H3K76me3 generation by up to eight times (Fig. 2f, g). To distinguish between these two scenarios, analysis of the H3K76 methylation status in *DOT1A* RNAi knock-down cells would be informative, as newly incorporated histone H3 would be tri-methylated at H3K76 by the end of G2 phase in the former case, but unmethylated in the latter case. However, this approach would be challenging because *DOT1A* knock-down cells are severely impaired[36].

In contrast to *DOT1A*, *DOT1B* is non-essential and can be knocked out, although DOT1B is the only H3K76me3 methyltransferase in *T. brucei* and although the majority of canonical histone H3K76 mod-ification is tri-methylated in *T. brucei*[18,35]. Indeed, *DOT1B* knock-out cells do not show a major phenotype, as only monoallelic VSG expression and differentiation are disrupted[18,35,39]. To date, the func-tion of H3K76me3 remains unknown. Curiously, both *DOT1B* and *his-tone H3.V* knock-out cells show a similar phenotype with defects in monoallelic VSG expression[35,39,64], raising the possibility that methyla-tion of H3.VK82, which corresponds to canonical H3K76, by DOT1B is involved in monoallelic VSG expression. Kinetoplastid-specific H3.V and H4.V are localized at transcription termination sites (TTSs) and sub-telomeric regions including VSG genes[20,33,64–66]. The majority of canonical histone H3K76 is tri-methylated, whereas more than half of variant histone H3.VK82 is mono-methylated (me1) with little me2 and ~30% me3 in vivo[20,31]. Between canonical histone H3 and variant histone H3.V, the sequence identity is 58%, and sequence similarity is only 72%. In particular, the DOT1A/B substrate H3K76- or H3.VK82-containing L1 loop sequences are highly dissimilar, and both substantially differ from the human H3 L1 loop sequence (*T. brucei* canonical H3: 72-SGAQ**K**[76]EGLRFQS-83, *T. brucei* variant H3.V: 78-VSNL**K**[82]DSYRMSA-89, and human/yeast/Xenopus H3: 75-AQDF**K**[79]TDLRFQS-86)[1,20]. As yeast genetic studies showed that mutations in the histone H3 L1 loop containing the H3K79 substrate would impact transcriptional silencing[67], it is plausible that the sequences adjacent to H3K76 in canonical H3 or H3.VK82 in variant H3.V would influence DOT1A/B

methyltransferase activities. It is not clear why H3.VK82 is mostly mono- and tri-methylated, nor how H3.VK82me1 and me2 are distributed in the *T. brucei* genome. Characterizing DOT1A and DOT1B methyltransferase activity toward variant nucleosomes would allow us to better understand DOT1A/B enzymology and DOT1B-dependent monoallelic VSG expression.

## Methods

### Expression and purification of DOT1 enzymes

Full-length *T. brucei* DOT1A (TriTrypDB[68] accession code Tb427_080022000) was amplified from genomic DNA of the *T. brucei* strain Lister 427 and cloned into a modified pFastBac1 vector (Invitrogen), resulting in pED459. All oligonucleotides used for cloning in this study are listed in Supplementary Table 2. The His$_6$-tagged DOT1A full-length protein was expressed in Sf9 insect cells, purified by a Ni-NTA agarose (Qiagen) column, followed by His$_6$-tag cleavage by PreScission protease (Cytiva), and further purified on anion exchange chromatography and gel filtration columns (Cytiva).

*T. brucei* Δ42-DOT1A (residues 43-295, pED464), *T. brucei* Δ27-DOT1B (TriTrypDB accession code of DOT1B: Tb427_010005600, residues 28-275, pED489), human DOT1L (UniProt[69]: Q8TEK3, residues 1–351 (pED608) and residues 1–420 (pED474)), *S. cerevisiae* Dot1p (UniProt: Q04089, residues 158–521, pED751), and their mutants were cloned into a modified pET28 vector. Δ42-DOT1A Arg254-to-Ala/Arg260-to-Ala double mutant (pED615), Δ42-DOT1A Glu281-to-Ser mutant (pED653), Δ42-DOT1A Arg105-to-Ala (pED661), Δ42-DOT1A Lys110-to-Gly (pED663), Δ42-DOT1A Asp247-to-Lys (pED682), Δ42-DOT1A Gly136-to-Arg/Gly138-to-Arg double mutant (pED752), Δ42-DOT1A Ser218-to-Ala (pED664), Δ42-DOT1A Phe246-to-Met (pED723), Δ27-DOT1B Ala197-to-Ser (pED665), Δ27-DOT1B Met225-to-Phe (pED724), DOT1L 1-420 Lys270-to-Asp (pED696), and DOT1L 1-420 Asn241-to-Ala (pED721) were generated using the QuikChange mutagenesis kit (Agilent Technology). Plasmids were transformed into LOBSTR-BL21(DE3)-RIL *E. coli* cells (Kerafast). Cells were grown in Luria-Bertani medium at 37 °C until the cell $OD_{600}$ reached ~0.6. Protein was expressed either by induction with 0.5 mM isopropyl-β-D-thiogalactoside (IPTG) at 18 °C for 16 h with a supplement of 0.1 mM zinc chloride or by autoinduction at 25 °C[70]. Cells were lysed by sonication and centrifuged at $35,000 \times g$ for 30 min. The clarified lysate was loaded onto a Ni-NTA agarose (Qiagen) column and eluted with imidazole. Protein-containing fractions were dialyzed in a buffer containing 20 mM Tris-HCl, pH 7.5, 50 mM NaCl, and 5 mM dithiothreitol (DTT) and subjected to His$_6$-tag cleavage with PreScission protease (Cytiva) at 4 °C for 16 h. The proteins were further purified by ion exchange (Cytiva) and gel filtration (GE Healthcare) columns.

### Histone purification

The *T. brucei* canonical histones H2A, H2B, H3, H4, and an N-terminal tail-less Δ18-H4 mutant (residues 19–99), human histones H2A, H2B, H3.1, H4, and a H2B K120C mutant were cloned into a modified pET28a, pET30a, or pET16b vector. Each plasmid was individually transformed into BL21(DE3) RIL-CodonPlus *E. coli* cells (Agilent). To produce *T. brucei* histone H3 with norleucine (Nle) in place of K76, a *T. brucei* H3(K76M) mutant was cloned into the vector pEW65 (pQE-81L H3(K79M), Addgene plasmid # 124100). The resulting plasmid was transformed into the methionine-auxotrophic *E. coli* strain B834(DE3), and the culture was grown in M9 minimal media supplemented with all amino acids to an $OD_{600}$ of ~0.6. Cells were pelleted, washed, and resuspended in fresh M9 media supplemented with all amino acids except methionine and supplemented with Nle. Protein overexpression of histones was induced with IPTG. Histones were purified from inclusion bodies as described[71,72]. Purified histones were flash-frozen and lyophilized. H2A, H2B, H3 or H3(K76Nle), and H4 or the N-terminal tail-less Δ18-H4 mutant were mixed at a ratio of 1.2:1.2:1:1 in unfolding buffer (7 M guanidine hydrochloride, 20 mM Tris-HCl, pH

7.5, and 10 mM DTT), and dialyzed against refolding buffer (10 mM Tris-HCl, pH 7.5, 2 M NaCl, 1 mM ethylenediaminetetraacetic acid (EDTA), and 5 mM β-mercaptoethanol (β-ME) at 4 °C for 16 h. After dialysis, the protein was concentrated and loaded onto a Superdex 200 (16/60) size-exclusion chromatography column (GE Healthcare) equilibrated in refolding buffer. The octamer peak fractions were concentrated to ~100 μM and used for nucleosome reconstitution.

### Preparation of ubiquitinated human histone H2B

His-TEV-Ubiquitin(G76C) (Addgene plasmid #75299) was expressed in BL21(DE3) CodonPlus *E. coli* cells (Agilent) using autoinduction at 25 °C and purified as described[9,70,73]. In total, 1.3 μmol of H2B K120C and 1.3 μmol His-TEV-Ub(G76C) were dissolved in 1 mL of 10 mM acetic acid and 7 M deionized urea. After resuspension, 9 mL of ligation buffer (50 mM sodium tetraborate, 6 M urea, and 5 mM Tris(2-carboxyethyl)phosphine (TCEP)) was added, and incubated for 30 min at 23 °C. Crosslinker (1,3-dichloroacetone, Sigma, cat#167479) equal to one-half molar ratio (~1.3 μmol) of total sulfhydryl groups (total SH- was 2.6 μmol) was diluted in N,N′-dimethyl-formamide (Sigma, cat# D4551) and added to the solution which was incubated on ice for an additional 30 min. The reaction was stopped by adding β-ME to a final concentration of 5 mM. The sample was diluted tenfold with denaturing binding buffer (50 mM sodium phosphate, 50 mM Tris-HCl pH 8.0, 300 mM NaCl, 6 M urea, 10 mM imidazole, and 5 mM β-ME), added to 10 mL of Ni-NTA beads, and mixed at 4 °C for 1 h. Ni-NTA beads were packed into a column and washed with binding buffer. Protein was eluted by a gradient of 10–400 mM imidazole. H2Bub1 was loaded on a SP cation exchange column (Cytiva), and H2Bub1-containing fractions were dialyzed against water, frozen, and lyophilized.

### DNA preparation

Widom 145 bp DNA sequence (5′-ATC GAT GTA TAT ATC TGA CAC GTG CCT GGA GAC TAG GGA GTA ATC CCC TTG GCG GTT AAA ACG CGG GGG ACA GCG CGT ACG TGC GTT AA GCG GTG CTA GAG CTG TCT ACG ACC AAT TGA GCG GCC TCG GCA CCG GGA TTC TGAT-3′) was used for the nucleosome assembly. A plasmid containing 12 copies of the 145-bp Widom 601 sequence separated by EcoRV restriction sites was produced in DH5α *E. coli* cells[74] and purified as described[71,72]. The plasmid was digested by EcoRV (NEB), and the plasmid backbone was precipitated by PEG 6000 (Sigma). The insert was further purified and concentrated by ethanol precipitation.

### Mononucleosome reconstitution

In total, 10 μM of histone octamers and 10 μM of 145-bp 601 Widom DNA were mixed in a 1:1.25 molar ratio in 2 M NaCl at 4 °C. This mixture was then dialyzed against buffers (20 mM Tris-HCl, pH 8.0, 1 mM EDTA, and 1 mM DTT) with gradually decreasing salt concentrations (1.5, 1.0, 0.6, and 0.25 M NaCl) at 4 °C, each for 3–16 h. A final dialysis was used to exchange buffer to 20 mM HEPES-NaOH pH 7.5, 1 mM EDTA, and 0.5 mM TCEP overnight at 4 °C. The nucleosome was concentrated to ~10 μM using a 50,000 MWCO Amicon Ultra concentrator (Millipore) and stored at 4 °C.

### Limited proteolysis

In a volume of 100 μL, full-length DOT1A at 2 mg/mL was incubated with a serial dilution of porcine elastase or chymotrypsin (Sigma) at 23 °C for 30 min. Reactions were stopped by adding reducing SDS-PAGE sample buffer and analyzed by 15% (w/v) SDS-PAGE.

### Crystallization, data collection, structure determination, and refinement

In total, 10–20 mg/mL of Δ42-DOT1A was incubated with twofold molar excess of AdoHcy for 1 h before crystallization. Crystals of Δ42-DOT1A in complex with AdoHcy grew in space groups $P2_12_12_1$ and $C222_1$ in hanging drops containing 1 μL of protein and 1 μL of mother liquor (10% (w/v) PEG 3000, 200 mM calcium acetate, and 100 mM

Tris-HCl, pH 7.0 for $P2_12_12_1$ and 30% (w/v) PEG 8000, 200 mM calcium acetate, and 100 mM MES-NaOH, pH 6.0 for $C222_1$). Crystals grew within 1 week at 20 °C, were cryoprotected by soaking in mother liquor supplemented with 25% (v/v) glycerol, and flash-cooled by plunging into liquid nitrogen. X-ray diffraction data were collected at beamline 9–2 at the Stanford Synchrotron Radiation Lightsource (SSRL). Diffraction data were processed in HKL3000[75]. The structure was solved by molecular replacement with Dot1p as the search model (PBD: 1U2Z[8]) using Phaser[76]. Model refinement was performed with Coot[77] and PHENIX[78] and assessed by MolProbity[79] (Supplementary Table 1). Molecular graphics were generated using PyMOL (Schrödinger, LLC).

## Methyltransferase assay

1 μM of nucleosome and 0.125 μM of DOT1 enzyme were incubated in 5.6 μM [methyl-$^3$H]-AdoMet (17.9 Ci/mmol; PerkinElmer)-containing methyltransferase buffer (50 mM Tris-HCl, pH 8.5, 50 mM NaCl, 1 mM EDTA, and 1 mM DTT). All enzymes were pre-incubated with AdoMet for 15 min before the addition of nucleosome at 25 °C. After 30 min at 25 °C, the reaction was stopped by adding 0.5% (w/v) SDS and precipitated by 10% (w/v) trichloroacetic acid (TCA) on ice for 15 min. The samples were blotted onto glass microfiber filters GF/F (GE Healthcare). The filters were dried, washed four times by cold 10% (w/v) TCA, and washed in cold acetone. Dried filters were subject to liquid-scintillation counting. Reactions were at least duplicated. Each radioactivity-based methyltransferase assay was performed with wild-type or no enzyme as control. Each wild-type data was aggregated into one data set, all wild-type and no-enzyme data in each figure is identical. The data were analyzed by one-way ANOVA followed by Tukey's post-hoc test using Prism 10 (GraphPad Software Inc.).

The radioactivity-based methyltransferase assay was conducted under co-factor-limited conditions (the 5.6 μM $^3$H-AdoMet concentration is around or below the $K_M$ value), where not all of the 2 μM H3K76me0 substrate is consumed. Product inhibition by AdoHcy further contributes that substrate is not fully consumed. By converting dpm to nanomolar concentration of transferred methyl groups for the 20 μL reaction volume, 533 dpm is equivalent to 1 nM of AdoMet. Therefore, 400k dpm is equivalent to 0.8 μM methyl groups being transferred from AdoMet to the nucleosome substrate. Overall, ~200–500 k dpm is equivalent to ~0.4–0.9 μM methyl groups being transferred at a H3K76me0 substrate concentration of 2 μM, indicating that substrate was not completely converted to product. Thus, the radioactivity-based methyltransferase assay served to assess if an enzyme was active or not. If active, the enzyme was subjected to the more quantitative MS-based assay for further characterization, where co-factor was not a limiting factor.

For quantitative MS, 0.125 μM of DOT1 enzyme and 1 μM of nucleosome was reacted in 160 μM AdoMet (NEB)-containing methyltransferase buffer (50 mM Tris-HCl, pH 8.5, 50 mM NaCl, 1 mM EDTA, and 1 mM DTT). The reaction was stopped by adding 1.6 mM AdoHcy at indicated time points, and the samples were stored at −20 °C.

## Histone methylation analysis by quantitative mass spectrometry (MS)

Histone H3 was prepared for chemical derivatization and digestion as described previously[80]. In brief, the lysine residues were derivatized with an anhydride reagent containing acetonitrile and propionic anhydride, and the solution pH was adjusted to 8.0 using ammonium hydroxide. The derivatized histones were then digested with trypsin in 50 mM ammonium bicarbonate buffer at 37 °C overnight. The N-termini of histone peptides were derivatized with the anhydride reagent, and the peptides were desalted with a self-packed C18 stage tip. The purified peptides were then dried in a centrifugal vacuum concentrator. The dried histone peptides were reconstituted in 0.1% formic acid. In all, 2 μL of the sample was injected onto the Acclaim PepMap 100 C18 column (3 μm × 0.075 mm × 150 mm) for the analysis

on a Q Exactive Plus instrument (Thermo Fisher Scientific) attached to a Vanquish Neo HPLC System (Thermo Fisher Scientific) and Nanospray Flex ion source (Thermo Fisher Scientific). The peptides were separated using buffer A (0.1% formic acid) and buffer B (80% acetonitrile and 0.1% formic acid) with a gradient of 2–35% over 50 min. The column was then washed at 98% buffer B over 5 min and equilibrated to 2% buffer B. Data-independent acquisition (DIA) was performed with the following settings. A full MS1 scan from 300 to 1100 $m/z$ was acquired with a resolution of 70,000, an automatic gain control (AGC) target of $1 \times 10^6$, and a maximum injection time of 50 ms. A series of DIA scans were acquired across the same mass range with sequential isolation windows of 24 $m/z$ with a normalized collision energy of 30, a resolution of 17,500, an AGC target of $2 \times 10^5$, and a maximum injection time of 60 ms.

The transition was built for the data analysis, and the peak was integrated using Skyline[81]. The data points for H3K76me0 and H3K76me1 were best fitted by a single exponential equation ($[C] = (C_0 - C_{Plateau})^* \exp(-k_{obs}^*t) + C_{Plateau}$, where $[C]$ is the H3K76me0 or H3K76me1 concentration, $k_{obs}$ is the observed rate constant, and $t$ is the reaction time) to follow the disappearance of H3K76me0 and H3K76me1 (GraphPad Prism 10 software) as described[82]. The $k_{obs}$ value for H3K76me1 disappearance was estimated from the 10 min time point onward (when nearly all H3K76me0 was converted to H3K76me1) for the DOT1A data set. DOT1B H3K76me0 disappearance and H3K76me3 appearance $k_{obs}$ values were also calculated.

## Fluorescence polarization assay

Fluorescence polarization measurements were carried out at 25 °C on a Victor$^3$V microplate reader (Perkin Elmer). The 6-carboxy-fluorescein (FAM)-labeled dsDNA probe (5 nM) was incubated for 10 min with increasing amounts of protein in 100 mM NaCl, 20 mM Tris-HCl, pH 7.5, 5% (v/v) glycerol, and 0.5 mM TCEP. No change in fluorescence intensity was observed with the addition of protein. The sequences of the oligonucleotides were FAM-5′-CTACAGTTCGTCAGGATTCC-3′ and 5′-CCGGAATCCTGACGAACTGTAG-3′. Curves were fit individually using GraphPad Prism 10 software (GraphPad Software, Inc.). Dissociation constants ($K_D$) were calculated as $[mP] = [maximum\ mP] \times [C] / (K_D + [C]) + [baseline\ mP]$, where $mP$ is milli polarization and $[C]$ is protein concentration. Averaged $K_D$ values and the standard error are reported as described[83].

## AdoMet cross-linking

AdoMet cross-linking was performed as described[8]. An aliquot of 2 μg of purified enzyme with 0.5 μCi of [methyl-$^3$H]AdoMet (17.9 Ci/mmol, PerkinElmer) in 20 μl of 50 mM BisTris propane (pH 8.5), 1 mM EDTA, and 0.5 mM DTT, were transferred to a 96-well plate on ice and placed 8 cm from an inverted UV transilluminator (Compact UV Lamp, UVGL-25 (Analytik Jena US)) using UV-C (254 nm wavelength) for 1 h. The protein was then separated by SDS-PAGE, stained with Coomassie, and subjected to fluorography. The gels were incubated in water for 30 min, then incubated in 5× volume of EN3HANCE solution (Perkin Elmer) for 1 h. Gels were washed in 1% glycerol for 1 h and dried. An X-ray film (Amersham Hyperfilm MP (Cytiva)) was exposed to the dried gel for 48 h at −80 °C, followed by development of the X-ray film.

## X-ray footprinting

For X-ray footprinting, nucleosomes were used that had H3K76 replaced by norleucine (Nle). Samples containing 5 μM nucleosome or nucleosome-Δ42-DOT1A (5 μM:15 μM) complex were exposed to hydroxyl radicals for intervals of 0, 12, 20 and 30 ms at ambient temperature at beamline 17-BM of the National Synchrotron Light Source II (NSLS-II) at Brookhaven National Laboratory (BNL)[84]. The samples were reduced with 10 mM DTT and alkylated with 25 mM iodoacetamide. Subsequently, two sets of samples were prepared including one set that was digested with Arg-C and AspN (Promega, Madison, WI),

# Article

and the second set that was digested with trypsin (Promega, Madison, WI). Sample digestions were performed at an enzyme-to-protein ratio of 1:10 (mol: mol) at 37 °C overnight. Then the two sets of digests were mixed at a 1:1 (v/v) ratio.

Identification and quantification of sites of oxidative modification were performed by liquid chromatography-MS (LC-MS) analysis using an Orbitrap Elite mass spectrometer (Thermo Electron, San Jose, CA) interfaced with a Waters nanoAcquity UPLC system (Waters, Taunton, MA). A total of 250 ng of proteolytic peptides were loaded on a trap column (180 μm × 20 mm packed with C18 Symmetry, 5 μm, 100 Å (Waters, Taunton, MA) to desalt and concentrate peptides, and subsequently eluted on a reverse phase column (75 μm × 250 mm nano column, packed with C18 BEH130, 1.7 μm, 130 Å (Waters, Taunton, MA) using a gradient of 2–42% mobile phase B (100% acetonitrile/0.1% formic acid) vs. mobile phase A (100% water/ 0.1% formic acid) over a period of 60 min at 40 °C with a flow rate of 300 nl/min. Peptides eluting from the column were introduced into the nano-electrospray source at a capillary voltage of 2.1 kV. All MS1 spectra of the eluted peptides were acquired in the Orbitrap (R = 120 K; AGC target = 400,000; MaxIT = auto; RF Lens = 30%; mass range = 350–1500) in the positive ion mode. MS2 spectra were collected in the linear ion trap (rate turbo; AGC target = 10,000; MaxIT = 35 ms; NCECID = 35%). The resulting MS/MS data were searched against *T. brucei* histones using Mass Matrix and Origin 8.0 software to identify sites of modification and to quantify modification rates, respectively. In particular, MS1 and MS/MS spectra were searched for peptides generated from *T. brucei* histone sequences by trypsin, Arg-C, and Asp-N digestions using mass accuracy values of 10 ppm and 0.7 Daltons for MS1 and MS/MS scans, respectively, with allowed variable modifications including carbamidomethylation for cysteines and all known oxidative modifications previously documented for amino acid side chains. In addition, MS/MS spectra for each site of proposed modification were manually examined and verified. Results from the free nucleosome was compared against the nucleosome-DOT1A mixtures. To normalize experimental variations between free nucleosome and nucleosome-DOT1A mixtures, we used the average of the mean and the median values for ratios measured in X-ray footprinting ($K_{\text{free}}/K_{\text{com}}$) across all peptides of a particular sample. For the nucleosome-Δ42-DOT1A complex, this value was 1.2. Hence, the ratios were divided by 1.2 to obtain the normalized ratios (NR). If the NR value for a given peptide is less than 1, it suggests that the corresponding region experienced gain in solvent accessibility due to structural changes introduced during complex formation. A NR value close to 1 indicates that the solvent accessibility of the region remains unchanged (no protection in the complex), while a NR > 1 suggests that the corresponding region exhibits protection from the solvent as a function of the complex formation. For the nucleosome-Δ42-DOT1A R254A/R260A mutant and nucleosome-Δ42-DOT1A D247K mutant mixtures, the average of the mean and median values of the ratios were close to 1 (0.97 and 0.97, respectively), and normalization was not applied. Because peptides of the nucleosome-Δ42-DOT1A complex showed significant changes (>2) in rate of modification with respect to the free nucleosome, these peptides were further analyzed for modifications for each individual amino acid residue within these peptides.

## Reporting summary

Further information on research design is available in the Nature Portfolio Reporting Summary linked to this article.

## Data availability

The X-ray structure coordinates and structure factors of *T. brucei* Δ42-DOT1A in complex with AdoHcy have been deposited in the Protein Data Bank (PDB) under accession codes 8FJN and 8FJM for the $C222_1$ and $P2_12_12_1$ crystal forms, respectively. Mass spectrometry data have been deposited in the MassIVE repository under accession code MSV000090680 [https://doi.org/10.25345/C54X54M8J] and in the ProteomeXchange database under accession code PXD038070 (linked to MassIVE). All other data generated in this study are available in the Source File provided with this paper. Source data are provided with this paper.

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

## Acknowledgements

We thank Tzanko Doukov [Stanford Synchrotron Radiation Light-source (SSRL)], Craig Ogata [Advanced Photon Source (APS)], and Erik Farquhar [National Synchrotron Light Source II (NSLS-II)] for support during data collection and the X-Ray Crystallography & Molecular Interactions Facility at the Sidney Kimmel Cancer Center, which is supported in part by National Cancer Institute Cancer Center Support Grant P30 CA56036 and S10 OD017987. Use of the SSRL, SLAC National Accelerator Laboratory, is supported by the U.S. Department of Energy (DOE), Office of Science, Office of Basic Energy Sciences under Contract No. DE-AC02-76SF00515. The SSRL Structural Molecular Biology Program is supported by the DOE Office of Biological and Environmental Research, and by the National Institutes of Health, National Institute of General Medical Sciences (including P41GM103393). GM/CA@APS has been funded by the National Cancer Institute (ACB-12002) and the National Institute of General Medical Sciences (AGM-12006, P30GM138396). This research used resources of the APS, a DOE Office of Science User Facility operated for the DOE Office of Science by Argonne National Laboratory under Contract No. DE-AC02-06CH11357. This research used resources of the NSLS-II, a DOE Office of Science User Facility operated for the DOE Office of Science by Brookhaven National Laboratory under Contract No. DE-SC0012704. pGEM-3z/601 was a gift from Jonathan Widom (Addgene plasmid # 26656). His-tagged yeast Dot1p (158–582) (plasmid pXC415) was a gift from Xiaodong Cheng. pEW65 (Addgene plasmid # 124100) and His6TEV-UbG76C (Addgene plasmid # 75299) were gifts from Cynthia Wolberger. J.K. and E.W.D. are supported by the NIH (R01 AI165840). B.A.G. is supported by the NIH (R01 AI165840 and R01 HD106051). We would like to thank the members of the Debler laboratory for helpful discussions and Christian Janzen for critical reading of the manuscript.

## Author contributions

V.S.F. performed protein purification, crystallization, X-ray data collection, structure determination, in vitro assays, and edited the manuscript; H.H. performed protein purification, structure data analysis, participated in experimental design and discussions throughout the study, and wrote the manuscript; T.N. and Z.Y. performed protein purification; Y.X., F.N.d.L.V., and J.B. performed mass spectrometry under the supervision of B.A.G.; J.K. performed X-ray footprinting; E.W.D. conceived the project, supervised the experimental work, and edited the manuscript. All authors were involved in reviewing the manuscript.

## Competing interests

The authors declare no competing interests.
