## [Peer Review File · Nature Communications]

Two DOT1 enzymes cooperatively mediate efficient ubiquitin-independent histone H3 lysine 76 tri-methylation in kinetoplastidsEditorial Note: Parts of this Peer Review File have been redacted as indicated to remove third-party material where no permission to publish could be obtained.

REVIEWER COMMENTS

Reviewer #1 (Remarks to the Author):

The manuscript by Frisbie and Hashimoto et al. well characterized two related histone H3 lysine methyltransferases from *Trypanosoma brucei* (Tb). These enzymes are eukaryotic homologs of yeast Dot1p and human DOT1L. However, Tb DOT1A and DOT1B are shorter in protein size (< 300 residues), and unlike Dot1p/DOT1L, their enzymatic activities are independent of H2B ubiquitination.

In the study, the authors used elegant quantitative mass spectrometry to monitor TbDOT1 activities on nucleosome substrates. They discovered that TbDOT1A prefers H3K76me0 as substrate, and generate me1/me2 products, while TbDOT1B prefers me1 as substrate and generates me2/me3 products. They quantitatively measured kinetic rate for each reaction step.

Next, they determine two crystal structures of TbDOT1A in complex with SAH, and found that the N-terminal domain folds into a Zn-coordinating beta-hairpin with extensive interactions with the methyltransferase domain. The structure allowed the authors to model a nucleosome-bound conformation and, using a series of mutational and biophysical method of X-ray footprinting, they found that (1) histone H4 N-terminal tail is not required for TbDOT1 activity, (2) TbDOT1 enzymes use unique motif X residues for activity, (3) DOT1-nucleosome interacting residues includes a conserved Arg finger for interaction with an acidic patch of nucleosome, (4) a unique K-to-E/D substitution in motif VI enhanced TbDOT1-nucleosome interaction, and (5) important residues in TbDOT1A and 1B for substrate specificity are identified.

Along the way, the authors also carried out parallel study on human DOT1L for comparison.

Overall, this is a very completed study. Some minor suggestions are below.

1. Abstract, the last sentence, "... cooperative action of two DOT1 enzymes in vivo" – are there any data showing these two enzymes have similar expression patterns in vivo?
2. Introduction, the end of first paragraph, besides of independent of H2B ubiquitination, are H3K76me1/2/3 in *Trypanosoma brucei* known to associate with any other histone modifications (H3K4me3, H4K16 acetylation)?
3. page 5, line 92, change "lack of cross-talk with ubiquitination" to "lack of cross-talk with H2B ubiquitination". Also, when the lysine is substituted to histidine, which is an amino acid that could also be modified post-translationally.
4. page 5, line 104, expand/explain the last sentence on why me3 generated by DOT1B cannot compensate for DOT1A defects in vivo. Does this suggest additional function of DOT1A in addition to its enzymatic activity?
5. page 7, lines 160-161, explain how the radioactive assay can tell DOT1A has a di-methyltransferase activity.
6. Discussion section contains results that should be moved to the Results section, e.g., DOT1A Ser218/DOT1B Ala196 swap mutations and DOT1L Asn241. Throughout the manuscript, the authors almost always have DOT1L as a comparison, which is good, but also it created difficulty to know which data panels are for DOT1L. Please consider organize all DOT1L data into a single figure.
6. Figure 3i, the labels are unclear: the WT and deltaN refer to nucleosome and H4 tail-less deletion. However, the same WT and deltaN could refer to the enzyme and N-terminal deletion. For the No enzyme control, what is the substrate present? Please go through every panel, make sure the labels are clear and meaningful enough without the need to read the text or legends.

Figure 4a, the motif alignment panels are truncated.

Figure 4c, some of the data points seem to have a single dot, no repeats?

Figure 5a, again, the sequence alignment is truncated, and what are the zinc coordinates? Is the same zinc atom bound by the N-terminal beta hairpin?

Figure 5b and 5e, the activity for the "WT" has a wide variation in panel e, this is also true in other figure panels. Please explain why.

Figure 6c, what is the meaning of the protection ratio? What are the numbers for the complete protection or no protection?

Reviewer #2 (Remarks to the Author):

The manuscript "Two DOT1 enzymes cooperatively mediate efficient ubiquitin² independent histone H3 lysine 76 tri-methylation in kinetoplastids" by Frisbie et al. addresses the mechanistic features of Dot1a and Dot1b's methyltransferase activity and substrate specificity. The authors show by kinetic analysis that Dot1a and Dot1b both need to efficiently generate H3K79 trimethylation on in vitro substrates. In their investigation, they also show that unlike Dot1L and Dot1 from *S. cerevisiae* Dot1a and Dot1b do not require H2B ubiquitination or H4 tail for H3K79 methylation to occur which is unique. Interestingly, they do go on to show that an acidic patch-arginine anchor interactions are used for substrate recognition for Dot1 methyltransferases. Although the in vitro data is interesting and well done, it would be interesting to tie some of their observations to new biological impact or insight.

An important aspect not fully addressed by the authors is why kinetoplastids developed two distinct Dot1 enzymes with differing substrate preference and product specificity? Some additional insight into this would be helpful, impactful and interesting. For example, the authors could do some detail protein phylogenetic analysis across many organisms/species to see if additional evolutionary insight can be obtained to help add biological reasoning behind their observed in vitro findings.

Additional issues:

Fig. 2b The authors should evaluate if there is any statistical difference between FL-Dot1A and the Delta42-Dot1A and Delta27-DOT1B. In addition, Fig 2b looks like the same data as shown in Figure 4C first three bar graphs. If so, the authors need to mention this either in the text or figure legend.

Fig 2c. The authors show that FL-DOT1A binds to DNA but show that it isn't required for methyltransferase (Fig. 2b). However, Fig 2b might suggest that DNA binding and/or the N-terminus (1-42) is inhibitory for activity. The authors could determine if this is the case. In addition, the authors could also investigate if the FL-DOT1A binds to RNA or RNA/DNA hybrids and if this alters chromatin association or activity. Thus, RNA binding or recognition of R-loops may also contribute to its association to chromatin or function as it does other chromatin associated factors.

I don't understand why the results stopped at figure 6 when there clearly are 7 figures? Or Why Figure 7 seemed to be part of the discussion? Was this intentional? Because of this, the discussion seemed disjointed where the result section is placed in the middle of the discussion. In addition, the histone H3V section needs further development and a bit strange to end on this note. For example, the authors should talk more about H3V and is it similar to other H3 variants in other species, etc. Furthermore, additional experimentation could make this section more relevant and interesting for the manuscript.

Re: Nature Communications manuscript NCOMMS-23-36050

We would like to thank the two referees for their positive comments about our work and appreciate their constructive suggestions for improving our manuscript. We have addressed all raised issues by carefully revising the text and the figures. As requested by both reviewers, we have moved Figure 7 to the Results part, and we have substantially expanded the Discussion in response to the reviewers' comments and questions to increase the biological insight and impact of our findings. Our detailed response to the reviewers' comments is outlined below and highlighted in red.

Point-by-point response:

Reviewer #1 (Remarks to the Author):

The manuscript by Frisbie and Hashimoto et al. well characterized two related histone H3 lysine methyltransferases from *Trypanosoma brucei* (Tb). These enzymes are eukaryotic homologs of yeast Dot1p and human DOT1L. However, Tb DOT1A and DOT1B are shorter in protein size (< 300 residues), and unlike Dot1p/DOT1L, their enzymatic activities are independent of H2B ubiquitination.

In the study, the authors used elegant quantitative mass spectrometry to monitor TbDOT1 activities on nucleosome substrates. They discovered that TbDOT1A prefers H3K76me0 as substrate, and generate me1/me2 products, while TbDOT1B prefers me1 as substrate and generates me2/me3 products. They quantitatively measured kinetic rate for each reaction step.

Next, they determine two crystal structures of TbDOT1A in complex with SAH, and found that the N-terminal domain folds into a Zn-coordinating beta-hairpin with extensive interactions with the methyltransferase domain. The structure allowed the authors to model a nucleosome-bound conformation and, using a series of mutational and biophysical method of X-ray footprinting, they found that (1) histone H4 N-terminal tail is not required for TbDOT1 activity, (2) TbDOT1 enzymes use unique motif X residues for activity, (3) DOT1-nucleosome interacting residues includes a conserved Arg finger for interaction with an acidic patch of nucleosome, (4) a unique K-to-E/D substitution in motif VI enhanced TbDOT1-nucleosome interaction, and (5) important residues in TbDOT1A and 1B for substrate specificity are identified.

Along the way, the authors also carried out parallel study on human DOT1L for comparison.

Overall, this is a very completed study. Some minor suggestions are below.

We thank the reviewer for the positive and critical feedback. We have addressed and incorporated the suggestions into our revised manuscript.

1. Abstract, the last sentence, "... cooperative action of two DOT1 enzymes in vivo" – are there any data showing these two enzymes have similar expression patterns in vivo?

RNA-seq-based mRNA abundance of *Tb*DOT1A and *Tb*DOT1B during the cell cycle has been analyzed by Archer *et al.*, 2011 PloS ONE (PMID: 21483801), while the protein expression levels have not been successfully analyzed yet (Janzen *et al.*, Mol Cell, 2006 PMID: 16916638). The mRNA expression data shown in the following figure (Left panel: DOT1A mRNA, <https://tritrypdb.org/tritrypdb/app/record/gene/Tb927.8.1920#ExpressionGraphs>, Cell Cycle Transcriptome. Right panel: DOT1B mRNA, <https://tritrypdb.org/tritrypdb/app/record/gene/Tb927.1.570#ExpressionGraphs>, Cell Cycle Transcriptome. TPM refers to Transcripts per Million) are accessible from TriTrypDB, a functional genomic database for trypanosomatids:

[redacted]

While DOT1B mRNA is about double the amount of DOT1A mRNA, both mRNAs are present at roughly constant levels throughout the cell cycle with a slight increase at the G2 phase. However, DOT1A/B protein abundance is currently not known due to the lack of specific antibodies, and it is unknown whether DOT1A and DOT1B are active throughout the cell cycle. Indeed, the existence of a regulatory cell cycle-dependent DOT1A/B methyltransferase activity mechanism has been proposed based on the observation that the DOT1A/B products H3K76me1 and me2 are only detected in G2/M but not in S phase (Janzen *et al.*, *Mol Cell* 2006 PMID: 16916638, and Gassen *et al.*, *NAR* 2012 PMID: 22941659). In the revised Discussion, we have included a section on DOT1A/B abundance and mention the recently published RNase H2-DOT1A/B interaction that may regulate DOT1A/B activity (see also response to point 4 below for more details).

2. Introduction, the end of first paragraph, besides of independent of H2B ubiquitination, are H3K76me1/2/3 in *Trypanosoma brucei* known to associate with any other histone modifications (H3K4me3, H4K16 acetylation)?

Human H3K79me2 is only present at 1~2% of chromatin (Sweet, *et al.*, 2010 JBC PMID: 20699226), enriched at transcription start sites (TSSs), and colocalized with H3K4me3, H4K16Ac, and H2Bub1. In *T. brucei*, by contrast, the majority of H3K76 (~80%) is tri-methylated, and H3K76me3 is widely distributed across the genome. H3K76me3 is not specifically enriched at TSSs, where H3K4me3, H4K10Ac, and the histone variant H2A.Z are enriched (Kraus *et al.*, *Nat. Commun.* 2020 PMID: 32198348, Wright, *et al.*, *Mol. Biochem. Parasitol.* 2010 PMID: 20347883,

and Siegel et al., *Genes Dev.* 2009 PMID: 19369410). So far, no modification has been reported or suggested to colocalize or be excluded with H3K76me1/2/3. In the revised manuscript, we have added this information in the Introduction to address *T. brucei* H3K76 methylation and other epigenetic modifications.

3. page 5, line 92, change “lack of cross-talk with ubiquitination” to “lack of cross-talk with H2B ubiquitination”.

Done.

Also, when the lysine is substituted to histidine, which is an amino acid that could also be modified post-translationally.

Previous analyses of histone PTMs in *T. brucei* (Janzen et al., *FEBS Letter* 2006 PMID: 16580668, Zhang et al., *iScience*. 2020 PMID: 32403088) have not identified any modification of H2B His108 (corresponding to human H2BK120). Unlike most eukaryote histone sequences, which are highly conserved, the *T. brucei* histone sequences have highly diverged. The *T. brucei* histone H2A amino acid sequence shares only 49%, H2B 41%, H3 56%, and H4 60% identity with human histones. As such, histone modifications are quite different from histones of most eukaryotes (Janzen et al., *FEBS Letter* 2006 PMID: 16580668, Figueiredo et al., *Nat. Rev. Microbiol.* 2009 PMID: 19528957, see figure below adapted from Figueiredo et al.). Therefore, it is impossible at present to assess how the Lys-to-His amino acid substitution in the *T. brucei* H2B sequence impacts its modification and its potential cross-talk with other posttranslational modifications.

[redacted]

In the revised manuscript, we have emphasized histone sequence divergence and expand on *T. brucei* epigenetic histone modifications in the Introduction and Discussion.

4. page 5, line 104, expand/explain the last sentence on why me3 generated by DOT1B cannot compensate for DOT1A defects in vivo.

This conundrum has been discussed since the discovery of DOT1A and DOT1B almost 20 years ago (Janzen et al., *Mol Cell* 2006 PMID: 16916638). Based on our DOT1A/B kinetics data, we can now provide an explanation for this finding. In short, *in vivo* data suggest that H3K76me1/me2 is important for cell cycle progression. Our analysis confirmed that DOT1B can methylate H3K76 from me0 to me3, but importantly revealed that me1 and me2 intermediates do not accumulate. Thus, if DOT1B can generate H3K76me3 from me0 *in vivo*, DOT1A knock-down cells would not accumulate H3K76me1/me2 and lead to cell cycle defects, even if DOT1B was active. Our *in vitro* data show that combination of DOT1A and DOT1B – concomitantly or sequentially – yields significant amounts of H3K76me1 and me2 and accelerates H3K76me3 generation by up to eight times. We have included this explanation in the revised Discussion.

Does this suggest additional function of DOT1A in addition to its enzymatic activity?

The DOT1A overexpression phenotype, in which cells contain >4n DNA content, is indeed dependent on methyltransferase activity, as overexpression of an AdoMet binding residue mutant (G138R) did not cause such a phenotype. By contrast, it is not known if the DOT1A knockdown, in which cells contain a 1n DNA content, is dependent on enzymatic activity (Gassen et al., 2012 *Nuc. Acids Res.* PMID: 22941659).

Recently, both DOT1A/B were found to form a complex with RNase H2 and PCNA *in vivo* (Eisenhuth et al., *mBio* 2021 PMID: 34749530 and Staneva et al., *Genome Res.* 2021 PMID: 34407985), suggesting a link of DOT1A/B with the replication machinery, as RNase H2 and PCNA are found at DNA replication foci in S phase. However, the DOT1A/B products H3K76me1/me2 are not detected in S phase, only in G2/M phase (Gassen et al., *NAR* 2012 PMID: 22941659). These findings indicate that RNase H2 may suppress or inactivate DOT1A/B H3K76 methyltransferase activity, which we confirmed *in vitro* (unpublished results). While further biochemical studies are required to address how DOT1A/B and RNase H2 interaction impact DOT1 methyltransferase activity and RNase H2 activity and how DOT1A-RNAi impacts DOT1A-RNase H2-PCNA complex and cell cycle progression, these studies are beyond the scope of this manuscript. In the revised manuscript, we have included the DOT1A/B and RNase H2 interaction in the discussion and discuss a potential regulatory role of RNase H2 for DOT1A/B activity.

5. page 7, lines 160-161, explain how the radioactive assay can tell DOT1A has a di-methyltransferase activity.

The radioactive assay can indeed not distinguish between mono-, di-, or tri-methyltransferase activity, and we have fixed this in the revised manuscript.

6. Discussion section contains results that should be moved to the Results section, e.g., DOT1A Ser218/DOT1B Ala196 swap mutations and DOT1L Asn241. Throughout the manuscript, the authors almost always have DOT1L as a comparison, which is good, but also it created difficulty to know which data panels are for DOT1L. Please consider organize all DOT1L data into a single figure.

We have revised the labelling in each panel for clarification.

7. Figure 3i, the labels are unclear: the WT and deltaN refer to nucleosome and H4 tail-less deletion. However, the same WT and deltaN could refer to the enzyme and N-terminal deletion. For the No enzyme control, what is the substrate present? Please go through every panel, make sure the labels are clear and meaningful enough without the need to read the text or legends.

We have revised the labelling for clarification.

8. Figure 4a, the motif alignment panels are truncated.

Fixed.

9. Figure 4c, some of the data points seem to have a single dot, no repeats?

We have done the repeat experiments, which further confirmed the results. Figure 4c with added statistical analysis was updated.

10. Figure 5a, again, the sequence alignment is truncated, and what are the zinc coordinates? Is the same zinc atom bound by the N-terminal beta hairpin?

We have updated the figure to clarify the Zn-coordinating cysteine residues.

11. Figure 5b and 5e, the activity for the “WT” has a wide variation in panel e, this is also true in other figure panels. Please explain why.

The relatively wide variation in Figure 5b and 5e (radioactivity-based assay) with respect to the smaller variation of the mass spec-based methyltransferase data can be explained by the different conditions of these two assays. In the radioactivity-based assay, we used 5.6 μM ^3H -AdoMet, 0.125 μM enzyme and 1 μM nucleosome, which corresponds to 2 μM of H3K76 residues. The total potential substrate amount in this assay is 4 μM (me0 and me1) for the di-methyltransferase DOT1A, and 6 μM (me0, me1, and me2) for the tri-methyltransferase DOT1B in a 20 μl reaction volume. Because the ^3H -AdoMet concentration 5.6 μM is around or below the K_M value, not all of the substrate can be consumed, unlike in the MS-based assay, which contains 160 μM AdoMet. In addition, product inhibition by AdoHcy entails that not all ^3H -AdoMet is used. By converting dpm to nanomolar concentration of transferred methyl groups in our 20 μL reaction condition, 533 dpm is equivalent to 1 nM of AdoMet. Therefore, 400k dpm is equivalent to 0.8 μM methyl groups being transferred from AdoMet to the nucleosome substrate in our 20 μL assay condition. While the absolute dpm numbers seem within a wide range (200-500k dpm), the concentration range of methyl groups transferred is only \sim 0.4-0.9 μM . This analysis shows that the radioactivity-based assay was conducted under co-factor-limited conditions and merely serves to assess if an

enzyme is active or not. If active, the enzyme was subjected to the more quantitative mass spec-based methyltransferase assay for further characterization, where co-factor was not a limiting factor. We have pointed this out in the revised Methods section.

12. Figure 6c, what is the meaning of the protection ratio? What are the numbers for the complete protection or no protection?

The protection ratios for each peptide are calculated as the fraction of modification rate K_{free} over K_{complex} , where “free” refers to the nucleosome, and “complex” refers to the complex of DOT1A and nucleosome. Upon complex formation, regions in the binding interface experience considerable loss in solvent accessibility (protection). Thus, the protection ratio measures the extent of decrease in oxidative modification in the peptide/s within the binding interface. A protection ratio value close to 1 indicates that the solvent accessibility of the region remains unchanged (no protection in the complex), while a value > 1 suggests that the corresponding region exhibits protection from the solvent as a function of the complex formation. Although the peptide can be completely protected from oxidative labeling in the complex ($K_{\text{com}} = 0$) against the free form of the protein, this event is rarely observed in footprinting experiments. Typically, in footprinting experiments, a decrease in modification rate for a specific peptide/s is observed upon complex formation with high protection ratios ≥ 2 and moderate protection ratios $\geq 1.5 < 2$. We have expanded the Methods section accordingly for clarification.

Reviewer #2 (Remarks to the Author):

The manuscript “Two DOT1 enzymes cooperatively 1 mediate efficient ubiquitin2 independent histone H3 lysine 76 tri-methylation in kinetoplastids” by Frisbie et al. addresses the mechanistic features of Dot1a and Dot1b’s methyltransferase activity and substrate specificity. The authors show by kinetic analysis that Dot1a and Dot1b are both need to efficiently generate H3K79 trimethylation on in vitro substrates. In their investigation, they also show that unlike Dot1L and Dot1 from *S. cerevisiae* Dot1a and Dot1b do not require H2B ubiquitination or H4 tail for H3K79 methylation to occur which is unique. Interestingly, they do go on to show that and acidic patch-arginine anchor interactions are used for substrate recognition for Dot1 methyltransferases. Although the in vitro data is interesting and well done, it would be interesting to tie some of their observations to new biological impact or insight.

We thank the reviewer for the positive feedback and for pointing out that our *in vitro* results would expand and accelerate *in vivo* studies in *T. brucei*.

1. An important aspect not fully addressed by the authors is why kinetoplastids developed two distinct Dot1 enzymes with differing substrate preference and product specificity? Some additional insight into this would be helpful, impactful and interesting. For example, the authors could do some detail protein phylogenetic analysis across many organisms/species to see if

additional evolutionary insight can be obtained to help add biological reasoning behind their observed in vitro findings.

Aravind et al. (Aravind, et al., *Prog Mol Biol Transl Sci* 2011 PMID: 21507350) performed a phylogenetic analysis of DOT1 and the results implied a lateral transfer of DOT1A and DOT1B from the animal lineage into mammalian *trypanosome* parasites. The distinct features of the N-terminal domain, including the Zinc-coordinating cysteines, were likely acquired after this transfer, followed by gene duplication that generated two functionally distinct paralogs. Since we fully agree with their plausible hypothesis, we have included their analysis in the Discussion of the revised manuscript.

Additional issues:

2. Fig. 2b The authors should evaluate if there is any statistical difference between FL-Dot1A and the Delta42-Dot1A and Delta27-DOT1B.

We have evaluated and added statistical differences using ANOVA for each radioactivity-based activity assay. Several points are important to note:

1) We used a different source of protein (Sf9 expressed full-length DOT1A vs. bacterially expressed Δ 42-DOT1A) and different proteins (Δ 42-DOT1A vs. Δ 27-DOT1B). Thus, different constructs and different batches of enzymes would show different activities, different purities, or different stabilities, which is commonly observed in such circumstances.

2) DOT1A and DOT1B showed distinct kinetics for me0 to me1, me1 to me2, or me2 to me3 (only in DOT1B), which the radioactivity-based assay cannot distinguish.

3) The radioactivity-based assay is co-factor-limited, see detailed response to reviewer #1 point 11, and, hence, H3K76 methylation only partially proceeds.

Given these limitations of the radioactivity-based assay, we utilize these data mainly to assess if an enzyme is active or not, while the mass spectrometry-based assay can quantitatively detect me0/me1/me2/me3 and provide detailed DOT1A/B enzyme kinetics.

In addition, Fig 2b looks like the same data as shown in Figure 4C first three bar graphs. If so, the authors need to mention this either in the text for figure legend.

We have clarified this in the figure legends.

3. Fig 2c. The authors show that FL-DOT1A binds to DNA but show that it isn't required for methyltransferase (Fig. 2b). However, Fig 2b might suggest that DNA binding and/or the N-terminus (1-42) is inhibitory for activity. The authors could determine if this is the case.

The response to this point is related to the response to the previous question #2. As the limitation of our radioactivity-based assay makes it difficult to discuss if the lower activity level is caused by inhibition by DNA binding and/or the N-terminal region (residues 1-42), we analyzed full-length (**Supplementary Fig. 4**) and $\Delta 42$ -DOT1A (**Figure 2d**) kinetics under more suitable conditions and using mass spectrometry for analysis. Importantly, both enzymes have comparable activity for the individual methylation steps. Therefore, it's difficult to conclude if full-length DOT1A activity is inhibited by the N-terminal region (residues 1-42), as the more suitable mass spectrometry-based assay did not show such inhibition.

In addition, the authors could also investigate if the FL-DOT1A binds to RNA or RNA/DNA hybrids and if this alters chromatin association or activity. Thus, RNA binding or recognition of R-loops may also contribute to its association to chromatin or function as it does other chromatin associated factors.

Human DOT1L and yeast Dot1p require a DNA binding region to enhance nucleosomal DNA interaction and decrease their K_M values toward both un-ubiquitinated and ubiquitinated nucleosomes (Worden et al., *Cell* 2019 PMID: 30765112). In this context, we tested if the Arg/Lys/His-enriched N-terminal region of DOT1A has a DNA binding affinity, as seen for Arg/Lys/His-enriched regions in DOT1L or Dot1p. As the DOT1A N-terminal region does indeed have a DNA binding affinity, albeit weaker ($\sim 10x$) than DOT1L or Dot1p, we named this region a "DNA binding region".

Recent reports about the DOT1A/B and RNase H2 interaction (Eisenhuth et al., *mBio* 2021 PMID: 34749530 and Staneva et al., *Genome Res.* 2021 PMID: 34407985) would raise the question if the DOT1A N-terminal tail has a possibility of binding DNA/RNA hybrid double strands or R-loops, which are substrates for RNase H2. Since positively-charged residues can interact with phosphate backbones, RNA/RNA, DNA/RNA, or R-loops, single or double strand nucleic acids could bind the N-terminal region of DOT1A. However, unlike the RNA/DNA hybrid binding domain (HBD) that is highly conserved across species (Nowotny et al., *EMBO J* 2008 PMID: 18337749), the N-terminal regions of DOT1A/B are not conserved, even among kinetoplastids. Additionally, the N-terminus of DOT1A is protease sensitive and would not form a stable domain structure, based on proteolysis results (**Supplementary Fig. 2**). We agree that the DNA binding region of DOT1A may bind RNA/DNA as well, which we now mention in the Results section, but it is unlikely to have DNA, RNA, or DNA/RNA hybrid specificity.

I don't understand why the results stopped at figure 6 when there clearly are 7 figures? Or Why Figure 7 seemed to be part of the discussion? Was this intentional? Because of this, the discussion seemed disjointed where the result section is placed in the middle of the discussion. In addition, the histone H3V section needs further development and a bit strange to end on this note. For example, the authors should talk more about H3V and is it similar to other H3 variants in other species, etc. Furthermore, additional experimentation could make this section more relevant and interesting for the manuscript.

After careful consideration, we have reorganized the Results and Discussion following reviewer's suggestion and made Figure 7 part of the Results. In general, we have completely revised and substantially expanded the Discussion, including the points mentioned above in response to the reviewers' comments and questions and including the section on histone H3V.

REVIEWERS' COMMENTS

Reviewer #1 (Remarks to the Author):

The authors addressed my questions/concerns. I have no further issues with the current revision.

Reviewer #2 (Remarks to the Author):

The authors have appropriately addressed my concerns and issues with their rebuttal letter and revised manuscript.